# Human-induced westerly jet shifts coordinate terrestrial productivity at the hemispheric scale

Xiaoye Yang [1], Aiguo Dai [2], Gabriele Messori [3,4,5], Bin He [6], Zhibo Li [1], Ziqian Zhong [7], Xing Yuan [8], Chang-Hoi Ho [9], Dim Coumou [10], Botao Zhou [11,12,13] & Deliang Chen [1,6,14] ✉

Previous studies have established how regional climate variability regulates local terrestrial gross primary productivity (GPP), yet the hemispheric-scale spatial organization of GPP, coordinated by large-scale atmospheric circulation, remains poorly understood. Here, using multi-source observations and numerical simulations, we show that anthropogenic shifts in Northern Hemisphere westerlies fundamentally reorganize terrestrial GPP patterns. Around 2000, westerly curvature reversed from a southward to a northward bend over eastern Europe, Northeast Asia, and western North America, while exhibiting opposite changes over central Asia and central North America. Spatial patterns of GPP trends during 1982–2018 closely match GPP responses to westerly curvature variations. Sensitivity analyses using CESM1 large-ensemble simulations and single-forcing experiments identify greenhouse gas forcing as the dominant driver of these changes, thereby reshaping GPP through surface climatic factors. Under the RCP8.5 scenario, continued curvature changes are projected to enhance GPP growth across northern Europe, Northeast Asia, and western North America, while suppressing productivity in southern Europe and central North America. These results reveal anthropogenic forcing influences terrestrial carbon uptake via large-scale atmospheric circulation, with important implications for predicting future carbon–climate feedback.

Terrestrial gross primary productivity (GPP)—the total carbon fixed by vegetation through photosynthesis—is a central component of the global carbon cycle and a critical mediator of climate–carbon feedbacks[1–5]. Over recent decades, satellite observations and Earth system model simulations have revealed a robust increase in global terrestrial GPP under anthropogenic warming, primarily driven by rising atmospheric $CO_2$ concentrations, longer growing seasons, and altered hydroclimatic conditions[6–9]. This overall increase, however, exhibits pronounced spatial heterogeneity[8], with certain hotspot regions—particularly in the Northern Hemisphere (NH) midlatitudes—experiencing substantially faster GPP gains than others[9–11]. Identifying

the drivers of this spatial divergence is essential for accurately projecting future land carbon uptake.

Extensive research has demonstrated that local climate factors, notably temperature and precipitation, strongly regulate GPP variability through well-established biophysical mechanisms[12–19]. Temperature influences GPP via multiple pathways: moderate warming within optimal ranges enhances photosynthetic enzyme activity, accelerates leaf development, and extends the growing season, thereby increasing carbon uptake[20–22]. In contrast, temperatures exceeding or falling below species-specific thresholds can suppress photosynthetic efficiency, increase respiration costs, or induce thermal and cold stress,

ultimately reducing GPP[22–27]. Precipitation further modulates GPP by controlling water availability through soil moisture and nutrient cycling, both of which are essential for vegetation growth[28,29]. Importantly, the relative importance of these climatic drivers varies regionally: GPP in energy-limited ecosystems is primarily temperature-controlled, whereas in water-limited regions it is more strongly governed by precipitation[29,30]. Using satellite-derived Normalized Difference Vegetation Index, a recent study has shown that terrestrial vegetation greening is regulated by both natural climate variability, such as the El Niño–Southern Oscillation, and anthropogenic climate change[31]. While local controls explain short-term and regional GPP variability, they struggle to fully account for the pronounced long-term spatial heterogeneity observed across the NH.

Notably, in some mid-latitude regions, the increase in GPP is significantly greater than in adjacent areas, even though these neighboring regions lie within a similar latitudinal range and thus experience comparable climatic conditions[2,32–34]. Large-scale atmospheric circulation—particularly the NH westerlies—plays a central role in shaping regional climate by modulating temperature and precipitation patterns and influencing extreme events through mechanisms such as Rossby wave propagation and the reorganization of pressure systems[35–38]. Under anthropogenic warming, a reduction in the equator-to-pole temperature gradient has contributed to weakened westerlies and summer storm tracks, with greenhouse gas (GHG) and aerosol forcing contributing at comparable levels[39]. Recent reanalysis-based studies also suggest an emerging poleward shift of the midlatitude jets, likely linked to tropical warming and broadly consistent with the range of historical climate model simulations[40]. At the same time, quasi-stationary wave patterns across the NH have intensified, associated with more frequent Greenland blocking, deep low-pressure systems over the eastern North Atlantic, and persistent high-pressure anomalies over Eastern Europe (EE)[27,28]. These circulation changes enhance the likelihood of concurrent extreme weather events by amplifying planetary-scale Rossby wave trains, thereby creating favorable conditions for spatially clustered heat and precipitation extremes[35–37,41–45].

Although numerous global-scale studies have documented trends in terrestrial GPP[46–48], most have attributed these changes primarily to local climate variations[22,28,49,50], largely overlooking the spatial organization of GPP as an integrated, system-level response. Previous work has demonstrated that large-scale atmospheric circulation, often influenced by oceanic and land surface conditions on interannual-decadal time scales[31], can generate synchronous, multi-regional extreme events and organize their spatial coherence, underscoring its hemispheric-scale influence[35,37,43–45]. Yet how such circulation-driven processes shape terrestrial GPP patterns remain largely unexplored. Moreover, the extent to which anthropogenic forcing drives circulation changes and their downstream impacts on GPP has not been systematically quantified. In this study, we introduce an image-processing-inspired framework to detect shifts in the curvature of NH westerlies and propose a dynamical mechanism by which large-scale circulation reorganizes GPP patterns. Leveraging large-ensemble climate model simulations, we further quantify the anthropogenic contribution to these circulation-driven changes in the terrestrial carbon uptake.

## Results

### Linking observed changes in westerly curvature to GPP responses

The large-scale upper-level circulation, particularly the NH westerlies, exhibit highly consistent patterns across multiple reanalysis datasets (Supplementary Fig. 1). During boreal summer (June–August), the strongest 200 hPa westerlies form a zonal belt between 35° N and 50° N. Climatologically, westerlies over Europe, East Asia (EA), and western and eastern North America are characterized by a southward trough, which correspond to negative curvature, whereas those extending

from EE to Central Asia (CA) and across central North America (CN) form a northward ridge, which have positive curvature (Fig. 1a and Supplementary Fig. 2). Over the past four decades, however, the curvature of the westerly jet at 200 hPa—a mathematically rigorous metric that captures streamline geometry and is widely interpretable across disciplines—has exhibited opposing trends across different longitudinal sectors, particularly over continental regions. Briefly, we computed the mathematical curvature at each point along the westerly jet axis at 200 hPa using ERA5 data (referred to as westerly curvature), which is defined by a series of longitude–latitude coordinates. It provides a quantitative measure of the local streamline geometry, with positive curvature indicating a northward ridge, and negative curvature a southward trough (see the "Indicator for westerlies curvature" section in Data and Method for methodological details).

Westerly curvature over EE, EA, and western North America (WN) has increased significantly, with a marked reversal around the year 2000: during 1979-1999, the westerlies exhibited a cyclonic southward dip, whereas during 2000–2023 they transitioned to an anticyclonic northward bulge (Fig. 1b–j). In contrast, CA and CN display significant decreasing trends in curvature (Fig. 1a, b). Based on these reversals, the NH is divided into eight longitudinal sub-regions: Western Europe (35° W–16.5° E), EE (16.5° E–47° E), CA (47° E–80.25° E), Eastern Asia (80.25° E–120.5° E), the Pacific Ocean (120.5° E–141.75° W), WN (141.75° W–106.75° W), CN (106.75° W–71.75° W), and Eastern North America (71.75° W–35° W).

Trends in summer-mean westerly curvature and summer mean of the absolute value of daily westerly curvature (referred to as absolute westerly curvature, a measure of deviation magnitude from zonal flow) during 1979–2023 reveal pronounced regional contrasts (Fig. 1c–j). EE and EA exhibit significant annual increases in curvature ($2.8 \times 10^{-5°}$ per year and $5.3 \times 10^{-5°}$ per year, respectively), while their absolute curvature remains unchanged, indicating a directional reversal from cyclonic to anticyclonic curvature without a change in overall bending intensity. By contrast, CA and CN show significant decreases in curvature ($-5.6 \times 10^{-5°}$ and $-3.2 \times 10^{-5°}$ $yr^{-1}$), again without significant changes in absolute curvature. Although parts of WN display localized increases, the regional mean trend is not statistically significant (Fig. 1h, k). Collectively, these results indicate a pronounced hemispheric-scale reversal in NH westerly curvature around 2000.

Consistent with these regional changes, overall westerly curvature —a measure of waviness of zonal flow (see Eq. (3) in "Data and Method") —has increased over the past four decades, reflecting enhanced meandering and greater spatial variability of westerly jet (Fig. 2a). This seems to support the notion that NH Jetstream may become wavier under global warming[51]. Regression analysis shows that years with higher overall curvature are characterized by a more pronounced northward bulge over EE, EA, and WN, accompanied by a deeper southward dip over CA and CN (Fig. 2b and Supplementary Fig. 3a). Importantly, correlations between summer GPP and westerly curvature remain robust after removing linear trends, indicating a strong interannual linkage that extends beyond long-term co-variability (Fig. 2c and Supplementary Fig. 3b). Given that anthropogenic land management can modulate or alleviate climate-driven constraints on productivity, we evaluated how land-use type affects the strength of the correlation between westerly jet curvature and GPP in the five hotspot regions shown in Fig. 2c. Each grid cell was classified as either managed land (croplands and urban areas) or natural land (primary and secondary forests) using LUH2 data, and the absolute correlation (|r|) between westerly jet curvature and GPP was calculated for each cell. The results show that the influence of large-scale westerly jet curvature on GPP is weaker over managed lands (Supplementary Fig. 4a), likely because human interventions such as irrigation, fertilization, and crop rotation buffer or override climate-driven variability[52,53]. In contrast, correlations are stronger in natural ecosystems (Supplementary Fig. 4b), where productivity is more directly constrained by climate.

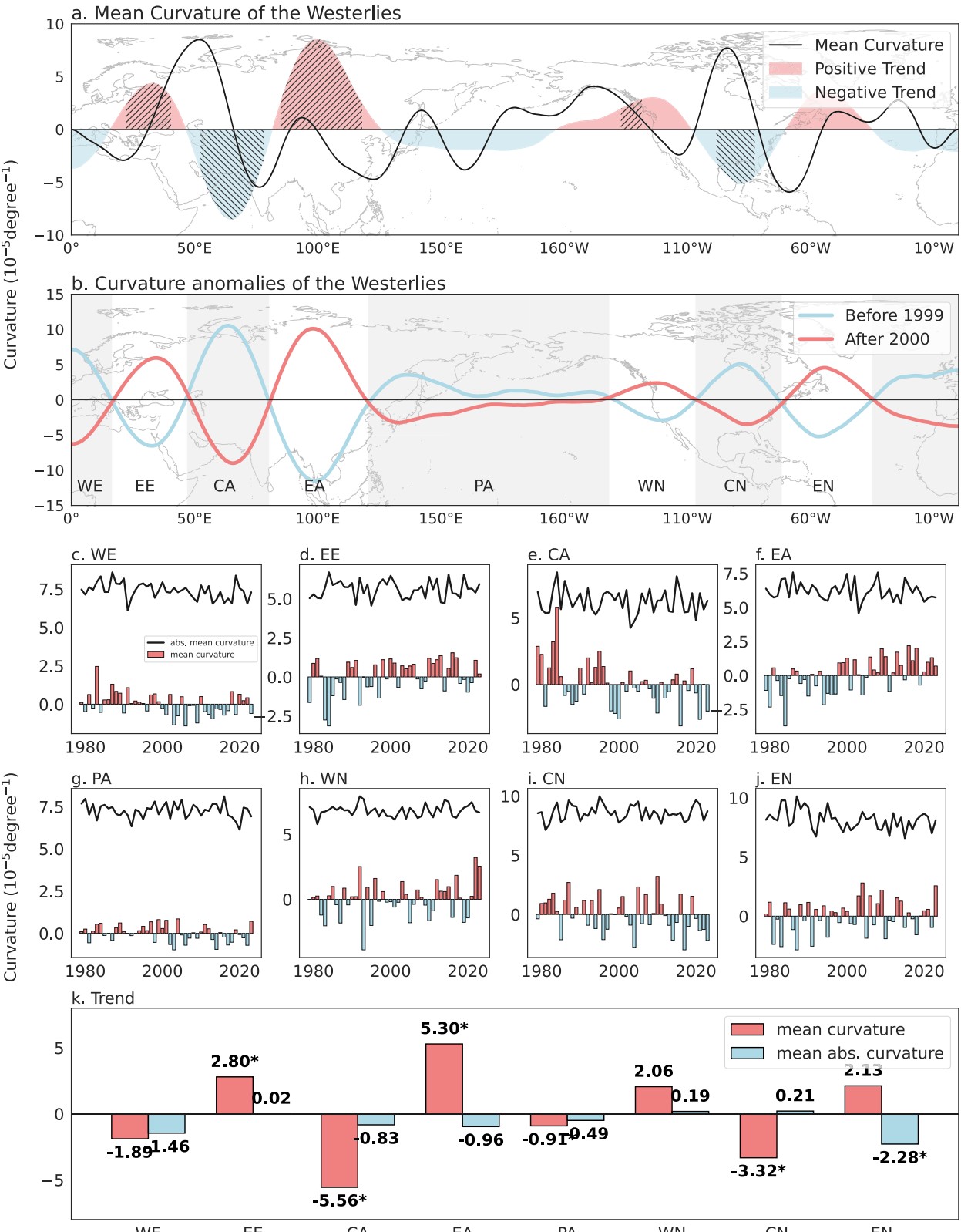

**Nature Communications** | (2026)17:4960

### Underlying mechanisms of GPP responses to westerly curvature changes

Changes in westerly curvature influence regional GPP through distinct atmospheric dynamical pathways (Fig. 3). Northward (anticyclonic) deflection of the westerlies (for a positive curvature) typically enhances incoming solar radiation and subsiding adiabatic motion, leading to higher surface temperatures, lower humidity, and reduced precipitation within the anticyclone region (Supplementary Fig. 5)[11,54–56]. Conversely, southward (cyclonic) deflection (for a negative curvature) promotes cloud formation, upward motion, increased precipitation, and suppressed solar radiation within the cyclone region[57]. The resulting GPP response is strongly region dependent as the westerly curvature varies regionally. Over Europe, for example, northward deflection suppresses GPP in the warmer south by

**Fig. 1 | Summer westerly curvature in the Northern Hemisphere and its changes during 1979–2023. a** Mean summer (June–July–August) curvature (black line, $10^{-5}$degree$^{-1}$) of the westerly jet axis at 200 hPa and its trend (shading, $10^{-5}$degree$^{-1}$year$^{-1}$) during 1979–2023 based on ERA5 daily data. Hatched areas indicate significant trends at the 95% confidence level. **b** Summer westerly curvature anomalies (relative to the 1979–2023 mean) during 1979–1999 (blue line) and 2000–2023 (red line). The Northern Hemisphere is divided into eight longitudinal regions: Western Europe (WE), Eastern Europe (EE), Central Asia (CA), Eastern Asia (EA), Pacific Ocean (PA), Western North America (WN), Central North America (CN), and Eastern North America (EN). The basemaps in (**a** and **b**) are shown only for

reference of the longitude range on the *x*-axis. The *y*-axis represents curvature and is unrelated to latitude. The westerly jet axis detection is performed within the Northern Hemisphere (0–90°N). A positive curvature indicates a northern ridge, while a negative curvature is for a southward trough. **c–j** Time series of summer mean westerly curvature κ (solid lines) and summer mean absolute westerly curvature |κ| (dashed lines) for **c** WE, **d** EE, **e** CA, **f** EA, **g** PA, **h** WN, **i** CN, and **j** EN. **k** Trends of summer mean westerly curvature (red bars) and summer mean absolute westerly curvature (blue bars) across the eight sub-regions. Asterisks indicate statistical significance at the 95% confidence level.

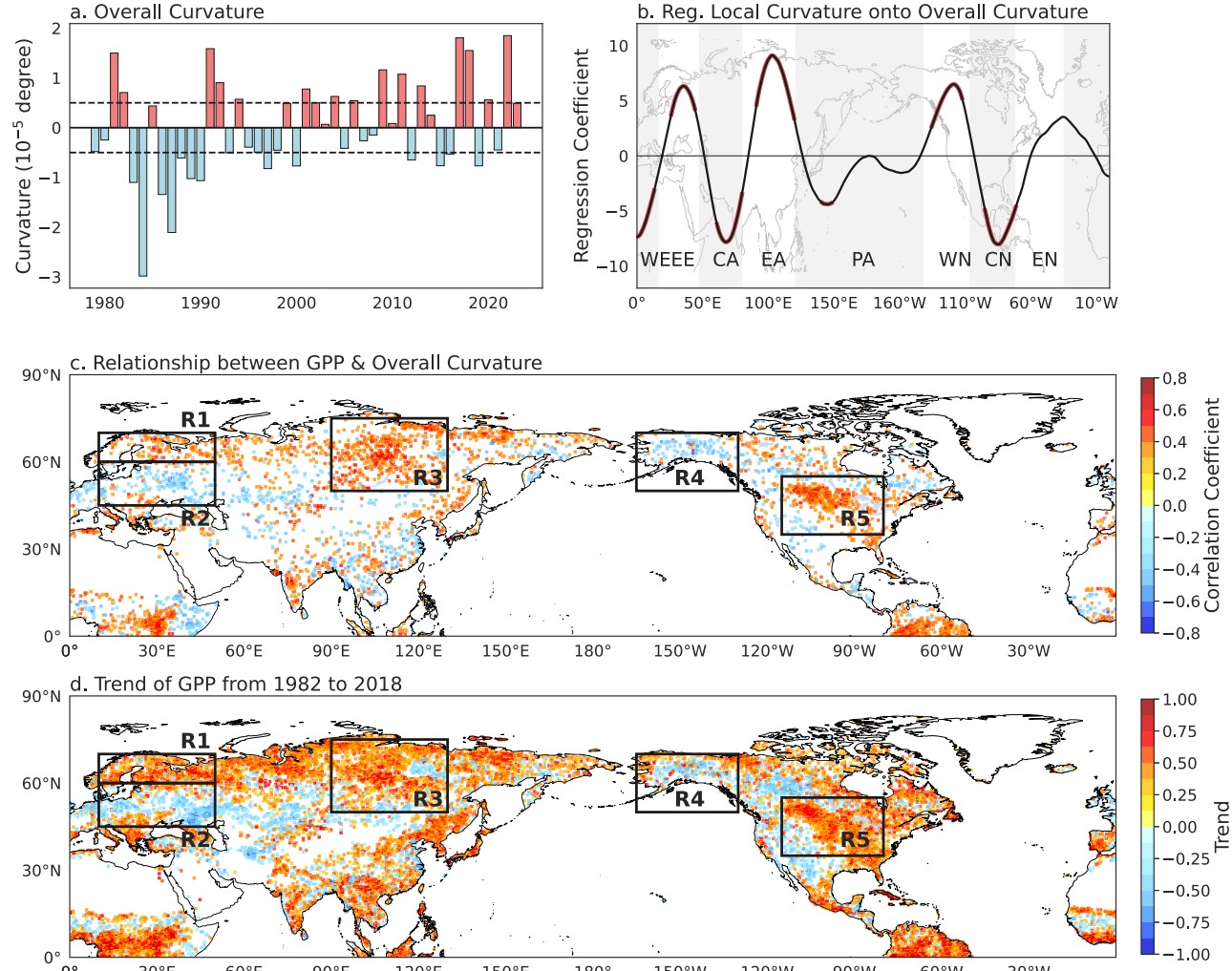

**Fig. 2 | Relationships between summer westerly curvature and gross primary productivity (GPP) in the Northern Hemisphere. a** Overall curvature anomaly (see Eq. (3) in "Data and Method", $10^{-5}$ degree) of summer westerlies in the Northern Hemisphere (1979–2023). Dashed lines indicate ±0.5 standard deviation. **b** Linear regression coefficients (unitless) of local westerly curvature onto overall curvature along each longitude. Red-shaded areas are significant at the 95% confidence level.

**c** Spatial distribution of the correlation coefficients between local summer GPP and Northern Hemisphere overall curvature during 1982–2018. Only correlations significant at the 95% confidence level are shown. **d** Trend of GPP (gCm$^{-2}$yr$^{-1}$) in the Northern Hemisphere during 1982–2018. Only statistically significant trends at the 95% confidence level are shown.

intensifying heat stress, but enhances GPP in the cooler north by alleviating temperature limitations. Although mean and maximum temperatures respond similarly to circulation changes, maximum temperature (Tmax) provides a more direct measure of the thermal extremes experienced by vegetation[20,23]. Such extremes often determine thresholds for photosynthetic inhibition, elevated respiration, and heat stress, thereby exerting a more immediate physiological constraint on GPP. A recent study also indirectly supports our findings from a different perspective[16], reporting that around the year 2000,

the temperature control on vegetation uptake of carbon in the NH midlatitudes underwent a shift, consistent with the timing of the westerly curvature changes identified in the present study. Additionally, observational datasets and large-ensemble simulations consistently capture these circulation–GPP linkages, revealing a coherent relationship between northward curvature, upper-level anticyclonic anomalies, and surface meteorological conditions (Figs. S5–7)[56,58].

Five hotspot regions (R1–R5) exhibiting strong curvature–GPP correlations coincide with sectors undergoing significant curvature

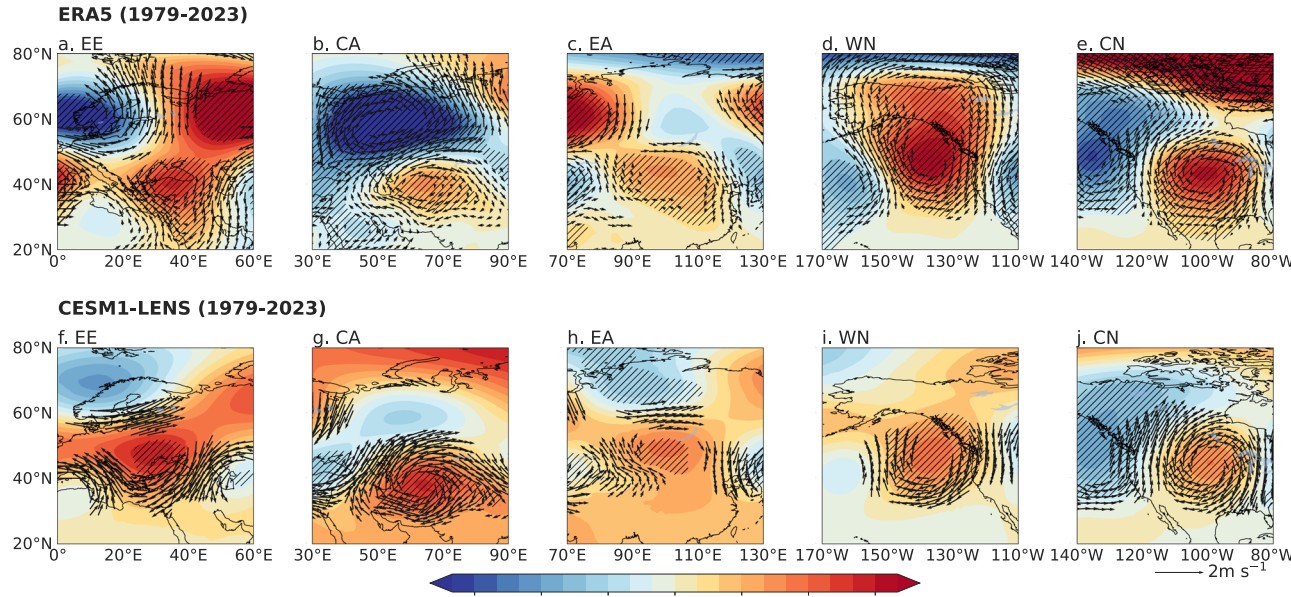

**Fig. 3 | Atmospheric circulation associated with westerly jet curvature. a–e** Linear regression coefficients of 500 hPa geopotential height (shading; hatching areas significant at the 95% confidence level) and 200 hPa horizontal winds (vectors; shown only where significant at the 95% level) regressed onto local westerly jet curvature in **a** Eastern Europe (EE), **b** Central Asia (CA), **c** Eastern Asia (EA), **d** Western North America (WN), and **e** Central North America (CN) during 1979–2023, based on ERA5 data. **f–j** same as (**a–e**), but derived from daily data from CESM1-LENS simulations. All analyses are performed on detrended data.

changes (Fig. 2c, d). GPP responses to circulation-induced temperature and precipitation anomalies are distinctly nonlinear (Supplementary Fig. 8 and S9). In the higher-latitude R1 region, GPP increases with Tmax up to approximately 19 °C before stabilizing, whereas in the central−southern R2 region, GPP declines once temperatures exceed optimal thresholds. Region R3 similarly shows enhanced GPP under moderate warming. In contrast, GPP in R4 is limited by reduced precipitation associated with anticyclonic circulation, while in R5, cyclonic curvature mitigates heat stress and stabilizes precipitation, supporting productivity. Together, these results demonstrate how large-scale atmospheric circulation interacts with regional climatic baselines to produce complex−and sometimes opposing−ecosystem response. In our analysis, large-scale variations in westerly circulation are conceptualized as a primary dynamical driver that structures regional climate and shapes the spatial distribution of GPP. Causal analysis further reveals that variations in westerly curvature may exert broad-scale influences on GPP through circulation adjustments; however, these effects are also mediated and amplified by local temperature and precipitation, which exert more immediate control over GPP. A detailed description of the causal analysis framework used to identify and quantify the direct and indirect influences of westerly curvature, circulation factors, and surface climate variables on GPP is provided in the Supporting Information (Supplementary Notes and Supplementary Fig. 10).

## Anthropogenic drivers of westerly curvature change

Over the past four decades, NH westerly curvature has undergone a pronounced change. To assess the role of anthropogenic forcing in this shift, we analyzed CESM large-ensemble simulations to quantify the response of westerly curvature to three individual forcings−GHG emissions, anthropogenic aerosols (AER), and biomass burning (BMB) −across five key longitudinal sectors: EE, CA, EA, WN, and CN. These results were compared with simulations forced by all external drivers (ALL) (Fig. 4). The model simulations successfully reproduce the observed curvature change and project a continued monotonic evolution of westerly curvature under the RCP8.5 scenario.

Under historical forcing, differences among the single-forcing experiments are small relative to internal ensemble variability. By contrast, under RCP8.5, clear divergence emerges between the all forcing simulations and the single-forcing experiments. In particular, simulations in which GHG forcing is held fixed exhibit pronounced departures from the ALL response, whereas simulations with fixed aerosol or BMB forcings closely track the all-forcing trajectory. This indicates that BMB exerts the weakest influence on westerly curvature, followed by AER, with GHG forcing playing the dominant role in driving curvature changes (Fig. 4f, g).

The observed patterns can be understood through established mechanisms, whereby GHG forcing modifies the meridional temperature gradient between the Arctic and mid-latitudes, thereby modulating the baroclinicity and wave propagation of the jet stream[59–61]. These changes in thermal-structure enhance jet meandering and amplify curvature anomalies, reflecting the link between Arctic amplification and shifts in mid-latitude circulation[38,62]. In addition, tropical warming provides an indirect pathway for influencing mid-latitude westerlies by reshaping upper-tropospheric waveguides and Rossby wave propagation, thereby complementing the effects of high-latitude warming on jet curvature[40,63].

Quantitative attribution further confirms this dominance. GHG forcing contributes $34.6 \pm 23.6\%$, $49.0 \pm 16.0\%$, $70.0 \pm 18.0\%$, $113.4 \pm 23.3\%$, and $93.5 \pm 16.1\%$ of the curvature change in EE, CA, EA, WN, and CN, respectively. In comparison, aerosol forcing contributes $-10.4 \pm 44.0\%$, $23.3 \pm 9.3\%$, $44.3 \pm 20.4\%$, $17.8 \pm 45.0\%$, and $23.6 \pm 24.9\%$ across the same regions. The upper-bound contributions exceeding 100% in North America indicate the presence of nonlinear interactions or additional suppressive processes affecting the net response. Consistent with these regional results, diagnostics of NH-wide curvature changes show a dominant GHG contribution ($100.6 \pm 14.6\%$), substantially exceeding that of AER ($21.0 \pm 20.3\%$). Overall, GHG forcing clearly governs the modeled changes in NH westerly curvature, whereas aerosol effects remain uncertain, particularly over EE and WN.

Building on this attribution, we examined how anthropogenic forcing reshapes terrestrial GPP patterns. Under GHG−driven warming,

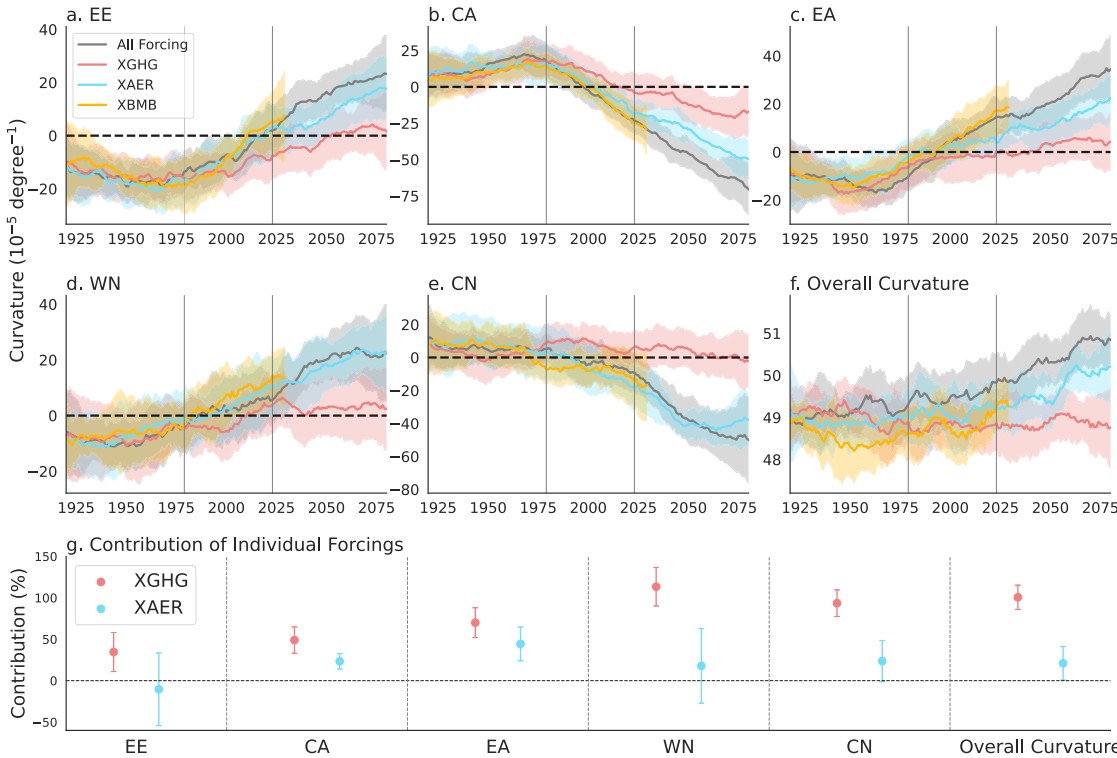

**Fig. 4 | Projected anthropogenic effects on westerly curvature changes based on CESM1-LENS. a–e** Time series of westerly curvature ($10^{-5}$ degree$^{-1}$) over **a** Eastern Europe (EE), **b** Central Asia (CA), **c** Eastern Asia (EA), **d** Western North America (WN), and **e** Central North America (CN) from 1920 to 2080. **f** Overall curvature ($10^{-5}$ degree$^{-1}$) of the summer westerlies in the Northern Hemisphere over the same period from different forcing runs by the CESM1. Solid lines denote multi-member ensemble means, and shaded areas indicate the inter-member spread corresponding to one standard deviation. **g** Contributions of greenhouse gas forcing (red) and aerosol forcing (blue) to changes in regional and overall Northern Hemisphere westerly curvature. Error bars represent the 95% confidence intervals. The sample sizes for groups All forcing, XGHG, XAER, and XBMB were 40, 20, 20, and 15, respectively.

NH GPP exhibits broadly coherent and statistically significant increases (Supplementary Fig. 11), reflecting both $CO_2$ fertilization and warming-induced stimulation of vegetation growth (Fig. 5)[11,14,64]. The strongest increases occur over Siberia, EA, and western and eastern North America. When GHG forcing is held constant, GPP continues to increase in many regions, but at a markedly reduced rate, with the largest gains confined to Western Europe and CN. Overall, the spatial pattern of the GPP trend difference between GHG and non-GHG forcing broadly resembles the observed pattern across most regions, although discrepancies remain in WN. In this region, the relationship between westerly curvature and GPP exhibits a distinct regional behavior. During the observational period, the negative GPP response to curvature is primarily associated with reduced precipitation (Supplementary Fig. 9h), which plays a dominant limiting role. In contrast, under GHG forcing, the projected increase in curvature is accompanied by a stronger rise in Tmax (Supplementary Fig. 12), which remains below the inflection point of the GPP response and thus favors vegetation productivity (Supplementary Fig. 8g). Meanwhile, changes in precipitation are comparatively weak due to the competing effects of circulation-induced subsidence and thermodynamically enhanced moisture availability. As a result, future changes in GPP are more strongly associated with temperature increases than with precipitation changes (Supplementary Fig. 13), leading to a divergence between observed trends and the GHG-minus-non-GHG signal in this region. This indicates that, although westerly circulation can systematically shape the large-scale spatial patterns of GPP across the NH, its impacts must be interpreted in the context of regional hydroclimatic differences.

Across the five hotspot regions, temperature increases drive GPP gains in R1 and R3, whereas in R2 and R5, excessive warming combined with reduced precipitation suppresses productivity. In R4, both rising

temperature and increased precipitation favor GPP enhancement. Together, these results demonstrate that the relationship between westerly curvature and GPP is nonlinear. When circulation-induced changes in meteorological conditions exceed physiological thresholds for photosynthesis, the curvature–GPP relationship can weaken or even reverse, as observed in CN.

## Discussion

Our study highlights the critical role of large-scale atmospheric circulation—particularly shifts in the curvature of NH westerlies—in shaping regional patterns of GPP under climate change. By linking these circulation changes directly to anthropogenic forcing, we provide a robust mechanistic explanation for the pronounced spatial disparities observed in GPP trends. The documented reversal in westerly curvature offers a clear, physically consistent framework for interpreting these variations, emphasizing the need to explicitly incorporate atmospheric dynamics into carbon cycle models and climate projections. CESM large-ensemble simulations further demonstrate that GHG emissions are the dominant driver of westerly curvature changes, which in turn govern regionally different GPP responses.

The influence of westerly curvature on GPP is strongly latitude-dependent, producing contrasting biospheric responses even under similar large-scale forcing. Anticyclonic curvature over EE, Northeast Asia, and WN intensifies planetary-wave activity and associated anticyclonic circulation anomalies, increasing the likelihood of extreme events such as heatwaves, droughts, and wildfires[43,44,65]. By contrast, cyclonic curvature over CA and parts of North America tends to offset warming, with corresponding effects on GPP and ecosystem stress[44,66,67]. These contrasting responses highlight how circulation-induced climate anomalies interact with regional climatic

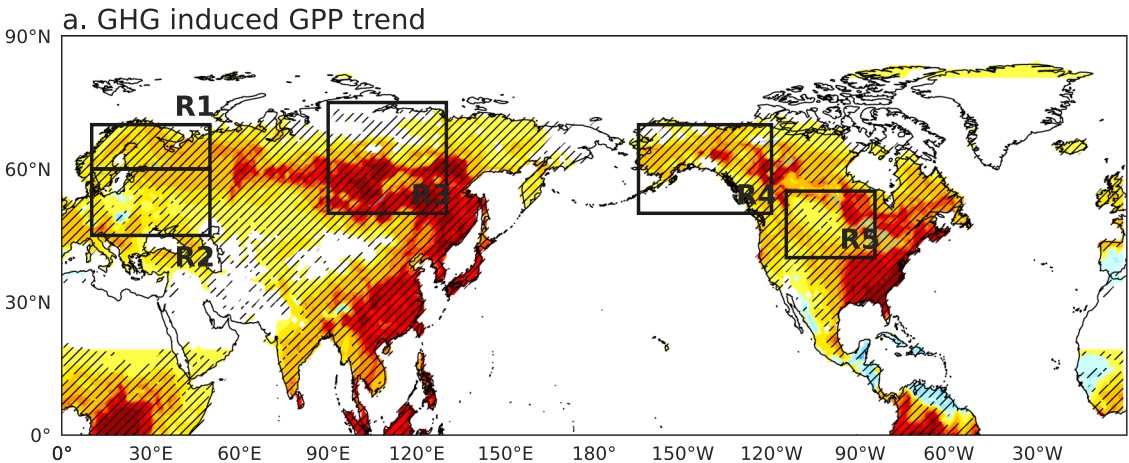

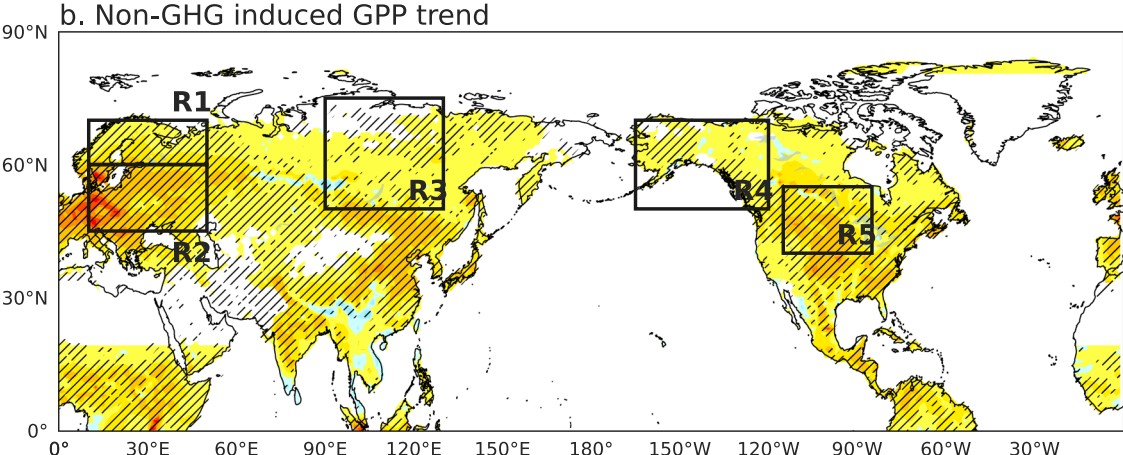

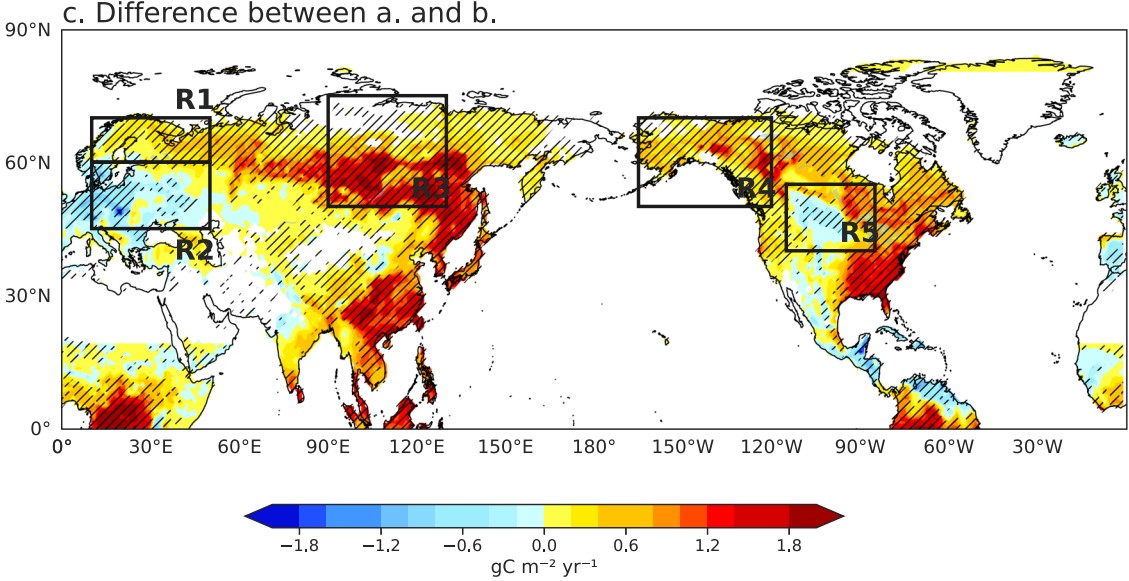

**Fig. 5 | Projected anthropogenic effects on Gross Primary Productivity (GPP) changes based on CESM1-LENS. a, b** Linear trends in summer GPP (g C m⁻² yr⁻¹) over the Northern Hemisphere during 2024–2080 under **a** greenhouse gas (GHG) forcing and **b** non-greenhouse gas (non-GHG) forcing. **c** Difference between (**a** and **b**), representing the net effect attributable to GHG forcing. Hatching areas significant at the 95% confidence level.

baselines and ecosystem sensitivities, thereby shaping ecosystem functioning.

Despite these advances, several key uncertainties warrant further investigation. First, finer-scale processes—including land–atmosphere feedbacks, vegetation composition, and soil moisture dynamics, as well as their reciprocal influences on atmospheric circulation—remain insufficiently constrained[65,68–70]. High-resolution regional climate–ecosystem models are needed to better resolve these coupled

processes. Second, the nonlinear and threshold-like responses of GPP to temperature and precipitation highlight the potential for tipping behavior, particularly in regions where climate conditions exceed optimal growth ranges[23,71]. Constraining these thresholds is critical for improving projections of future carbon cycle dynamics.

These findings have important implications for terrestrial carbon sequestration. Regions experiencing enhanced westerly curvature, including Northern Europe, Northeast Asia, and WN, are likely to function as increasingly effective carbon sinks in coming decades. In contrast, areas with stagnating or declining GPP growth—such as Southern Europe and CN—may face reduced carbon sequestration capacity. This spatial heterogeneity underscores the importance of realistically representing atmospheric circulation dynamics in Earth system models to reduce uncertainties in carbon budget assessments.

From a practical perspective, our results provide guidance for regional ecosystem management and climate mitigation strategies. Identifying regions with increasing GPP potential can inform afforestation, reforestation, and land-use planning aimed at maximizing carbon storage, while regions experiencing declining productivity may require adaptive management to prevent ecosystem degradation and carbon loss. More broadly, our findings establish large-scale atmospheric circulation as a key intermediary linking anthropogenic forcing to biospheric responses. Overall, this study advances understanding of GPP dynamics, elucidates the biospheric consequences of anthropogenic climate forcing, and provides critical insights for climate policy and mitigation strategies aimed at sustaining global carbon stocks.

## Methods
### Reanalysis and observation datasets
We used daily 200 hPa meridional (V, m·s$^{-1}$) and zonal (U, m·s$^{-1}$) wind, and geopotential height (Z, gpm) data on 0.25° grids from 1979 to 2023 from the ERA5 reanalysis[72], provided by the European Centre for Medium-Range Weather Forecasts. Daily maximum surface air temperature and precipitation data on 0.5° grids for the same period were obtained from the Climate Prediction Center (CPC), which were interpolated from station observations by CPC.

Vegetation productivity was represented by monthly global terrestrial GPP estimates derived from satellite-based near-infrared reflectance of vegetation (NIRv), spanning 1982–2018 at 0.05° resolution[73]. This GPP dataset captures seasonal and interannual variability effectively, supporting its use for estimating land–atmosphere carbon fluxes and informing projections under future climate scenarios[73–76]. To evaluate whether the robustness of GPP responses to westerly jet curvature is affected by dataset differences, we additionally used three GPP products for validation (the additional GPP data are used Supplementary Fig. 14): (1) GLASS GPP, covering 1982–2018 at 0.05° resolution, derived using an eddy covariance–light use efficiency model that integrates multiple globally used LUE models[77–79]. (2) LUE GPP, a global monthly GPP product covering 1982–2016 at 8 km resolution, estimated with an optimized LUE model based on FLUXNET data, combined with GIMMS3g FPAR and MERRA-2 meteorology data[80]. (3) LRF GPP, spanning 1982–2016 at 0.05° resolution, estimated using an asymptotic light response function between GPP and incoming photosynthetically active radiation data[81]. The results show that while minor spatial differences exist among datasets (e.g., in tropical Africa and eastern China), the overall spatial pattern in mid- to high-latitude regions is highly consistent, particularly within the hotspots considered in this study (Supplementary Fig. 14). In addition, except for the LUE GPP, the other GPP products show a high degree of consistency in the spatial distribution of statistically significant responses to westerly curvature, reaching an average agreement of 78.3% within the hotspot regions (using the NIRv GPP dataset as a reference). Although the LUE GPP exhibits a similar spatial pattern, it yields fewer statistically significant pixels

(Supplementary Fig. 14b). These differences likely reflect structural uncertainties among GPP products arising from different model assumptions, input datasets, and parameterizations, particularly in light-use-efficiency-based estimates. Overall, the main conclusions of this study are robust and not sensitive to the selection of GPP dataset.

LUH2 provides annually resolved, spatially explicit information on global land-use states and transitions at a horizontal resolution of 0.25°, including cropland, urban, forest, and other land-cover types[82,83]. This dataset enables us to distinguish intensively managed croplands from natural ecosystems in our analyses and to better account for potential human influences on GPP variability. We used LUH2 to calculate mean land-use fractions over the historical period 1982–2015.

To ensure the robustness of our analysis, we also used relevant variables from the NCEP–NCAR Reanalysis 1[84] and JRA-55[85] datasets to verify westerly changes across the NH. By incorporating multiple independent datasets, we minimize reliance on any single product and strengthen the reliability of our results.

### CESM1 large ensemble simulations
We employed the Community Earth System Model (Version 1) Large Ensemble (CESM1-LENS) developed by the National Center for Atmospheric Research (NCAR), a fully coupled Earth system model comprising 40 ensemble members designed to explore long-term climate change and projection uncertainties[86]. The ensemble is forced with all historical forcings from 1920 to 2005, followed by the RCP8.5 scenario from 2006 to 2100, which assumes a radiative forcing of 8.5 W m$^{-2}$ by the end of the 21st century. Each member is initialized with small, round-off level perturbations in the atmospheric temperature field, while sharing identical external radiative forcing.

To isolate the effects of individual anthropogenic forcings, we also used the CESM Single-Forcing Large Ensemble Project (CESM-XLENS), widely applied in climate change attribution studies[87,88]. Our analysis includes three configurations: the fixed-aerosol experiment (XAER; 1920–2080, 20 members), the fixed-GHG experiment (XGHG; 1920–2080, 20 members), and the fixed-biomass burning experiment (XBMB; 1920–2029, 15 members). In each, the target forcing is held constant at its 1920 level while all other forcings evolve, allowing the isolation of that forcing's contribution to future climate changes.

From these simulations, we obtained daily 20 hPa zonal and meridional winds (U200 and V200), daily maximum surface air temperature (TREFHTMX), precipitation (PRECET), and monthly GPP. Multi-member ensemble means were used to represent forced changes, while differences among members capture internal climate variability. These simulations enable the quantification of the linear impact of individual forcings on changes in the westerlies and their downstream effects on GPP:

$$\begin{cases} \Delta C_{GHG} = \frac{(C_{ALL} - C_{XGHG})}{C_{ALL}} \times 100\% \\ \Delta C_{AER} = \frac{(C_{ALL} - C_{XAER})}{C_{ALL}} \times 100\% \\ \Delta C_{BMB} = \frac{(C_{ALL} - C_{XBMB})}{C_{ALL}} \times 100\% \end{cases} \quad (1)$$

In Eq. (1), $\Delta C_{GHG}$, $\Delta C_{AER}$, and $\Delta C_{BMB}$ represent the contribution of GHG emissions, aerosol emissions, and BMB emissions, respectively, to changes in variable C (e.g., surface air temperature, precipitation, GPP). $C_{ALL}$ refers to variable $C$ in the all-forcing experiment, while $C_{XGHG}$, $C_{XAER}$, and $C_{XBMB}$ denote variable $C$ in the fixed GHG, aerosol, and BMB forcing experiment, respectively.

### Detection of westerlies changes in the Northern Hemisphere
NH westerly belt exhibits curvature—either northward or southward—across different longitudinal sectors, influenced jointly by internal atmospheric variability and external forcings[89,90]. Previous studies

have proposed various metrics to quantify atmospheric waviness[91–93]. For instance, some metrics assess waviness based on the length of specific geopotential height contours, while others derive meridional circulation indices from the zonal wind contribution[94–96]. These approaches primarily capture overall variations in NH westerlies, offering insight into large-scale circulation changes and contributing to debates on whether a reduced poleward temperature gradient under global warming enhances jet stream meandering and atmospheric blocking frequency[36,97].

However, existing metrics often cannot resolve regional directional shifts in westerly patterns. Enhanced troughs and ridges may both indicate increased waviness, yet they have fundamentally different implications for regional climate[98]. To address this limitation, we introduce an image-processing-inspired method to systematically detect and characterize changes in the NH westerlies.

We first applied morphological operations to daily 200 hPa zonal wind fields over the NH (0–90°N). A "closing operation" sharpened the westerly core structures by smoothing weaker branches within high-speed channels, enhancing the coherence of the main flow. This was followed by an "opening operation" to remove isolated high-wind anomalies associated with turbulence outside the primary westerlies. The primary westerly axis was then identified as the latitude of maximum wind speed.

Morphological filtering differs from conventional spatial filters: instead of using arithmetic operations such as convolution or Fourier transforms, it employs a structuring element as a kernel and computes center pixel values using set-theoretic operations over the neighborhood. This preserves geometric structures while effectively removing noise. Such methods have been widely applied in atmospheric research for identifying cloud morphology and structural patterns in satellite imagery[99,100]. Detailed mathematical principles are provided in previous studies[101,102].

Supplementary Fig. 15 illustrates the workflow for detecting the westerly axis, while Supplementary Fig. 16 shows the annual mean 200 hPa wind and the detected westerly axis from 1979 to 2023. The identified axis closely follows the zonal distribution of maximum wind speeds, confirming the reliability of the method. Importantly, the methodology—and the conclusions drawn—are robust across datasets. Supplementary Fig. 1 compares westerly variations detected using NCEP I and JRA-55 reanalysis with ERA5, demonstrating a high degree of consistency and reinforcing the robustness of our approach.

### Indicator for westerlies curvature

In this study, the westerly jet axis is represented as a series of latitude–longitude points: $([(x_1,y_1), (x_2,y_2),..., (x_n,y_n)])$. To quantitatively characterize the degree of bending of the westerly flow at each longitude, we employ the mathematical concept of curvature of the westerly jet axis. Compared with other dynamical metrics, curvature provides an intuitive, concise description of westerly flow patterns and facilitates the assessment of continuous variations along each longitude.

The curvature of the westerly jet axis is defined as:

$$\kappa = \frac{\frac{d^2y}{dx^2}}{\left(1 + \left(\frac{dy}{dx}\right)^2\right)^{\frac{3}{2}}} \quad (2)$$

$\kappa$ denotes the curvature at a given longitude $x$ and latitude $y$. Derivatives are approximated using a central difference scheme, which computes changes based on neighboring points. A positive curvature indicates a northward bulge (atmospheric ridge), whereas a negative curvature indicates a southward dip (atmospheric trough). The absolute westerly curvature, $|\kappa|$, quantifies the degree of bending regardless of direction; higher values reflect stronger deviations from a pure

zonal flow; it should be noted that here "deviation" refers to the amplitude of curvature rather than horizontal displacement of the jet axis.

For an intuitive interpretation: a curvature of $\kappa = 1$ degree$^{-1}$ implies that the flow direction changes by 1° per 1° of longitude traveled. Importantly, this describes the change in flow direction along the streamline, not a literal change in latitude per longitude.

Prior to curvature calculation, the westerly axis is smoothed using a window of ~1000 km to remove mesoscale disturbances and prevent abrupt variations between adjacent grid points. The trend of curvature variations remains robust to changes in the smoothing window length across different regions.

To assess the overall meandering of the westerlies across the NH, we calculate the root mean square of curvature across all longitudes:

$$Overall\ Curvature = \sqrt{\frac{1}{N}\sum_{i=1}^{N}\kappa_i^2} \quad (3)$$

where $K_i$ is the curvature at each longitude point and $N$ is the total number of points. A smaller overall curvature indicates a more zonal, less meandering westerly jet.

## Data availability

All data used in this study are publicly available. The ERA5 reanalysis dataset can be obtained from https://cds.climate.copernicus.eu/datasets. Daily maximum surface air temperature and precipitation data from the CPC can be obtained from https://psl.noaa.gov/data/gridded/index.html. Monthly global terrestrial GPP (NIRv) can be obtained from https://figshare.com/articles/dataset/Long-term_1982-2018_global_gross_primary_production_dataset_based_on_NIRv/12981977/2. GLASS GPP, LUE GPP, and LRF GPP dataset can be obtained from https://glass.hku.hk/download.html, https://catalog.data.gov/dataset/global-monthly-gpp-from-an-improved-light-use-efficiency-model-1982-2016-a36e3?utm_source=chatgpt.com, and https://doi.org/10.17894/ucph.b2d7ebfb-c69c-4c97-bee7-562edde5ce66, respectively. NCEP-NCAR Reanalysis 1, and JRA-55 datasets can be obtained from https://psl.noaa.gov/data/gridded/data.ncep.reanalysis.html and https://climatedataguide.ucar.edu/climate-data/jra-55, respectively. The CESM1-LENS are available at CESM Large Ensemble Community Project (https://www.cesm.ucar.edu/community-projects/lens), and CESM-XLENS can be obtained from the CESM1 Single Forcing Large Ensemble Project (https://www.cesm.ucar.edu/working-groups/climate/simulations/cesm1-single-forcing-le).

## Code availability

Image processing was conducted using functions from the *scipy.ndimage* package in Python (https://docs.scipy.org/doc/scipy/reference/ndimage.html). LK information flow can be computed using the Python package LK-Info-Flow (https://pypi.org/project/LK-Info-Flow/). The code for the analysis and mapping can be obtained via GitHub at https://github.com/xiaoyeyang1024/WesterlyJetAxis (ref. 103).

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

## Acknowledgements

D. Chen is supported by Carbon Neutrality and Energy System Transformation (CNEST) Program and Tsinghua University (100008001) and the Swedish Research Council (2021-02163 and 2022-06011). Z. Zhong was supported by the VAPOR project (101154385), funded by the Horizon Europe, MSCA Postdoctoral Fellowships 2023. D. Coumou was supported by the European Union's Horizon 2020 Research and Innovation Program (EXPECT, grant agreement no.101137656). G. Messori was supported by the Swedish Research Council Vetenskapsrådet (2022-06599, 2025-04998) and by the Swedish Research Council FORMAS (2025-04998).

## Author contributions

X.Y. designed and conducted the study and drafted the initial manuscript. A.D., G.M., B.H., Z.L., Z.Z., X.Y., C.-H.H., D. Coumou, B.Z., and D. Chen contributed substantially to data interpretation, manuscript revision, and refinement of the analysis framework. All authors provided critical feedback, contributed to improving the scientific clarity and robustness of the study, and approved the final version of the manuscript.

## Funding

## Competing interests

The authors declare no competing interests.

## Additional information

¹Regional Climate Group, Department of Earth Sciences, University of Gothenburg, Gothenburg, Sweden. ²Department of Atmospheric and Environmental Sciences, University at Albany, State University of New York, Albany, NY, USA. ³Department of Earth Sciences, Uppsala University, Uppsala, Sweden. ⁴Swedish Centre for Impacts of Climate Extremes (climes), Uppsala University, Uppsala, Sweden. ⁵Department of Meteorology, Stockholm University, Stockholm, Sweden. ⁶Department of Earth System Science, Tsinghua University, Beijing, China. ⁷Department of Space, Earth and Environment, Division of Geoscience and Remote Sensing, Chalmers University of Technology, Gothenburg, Sweden. ⁸State Key Laboratory of Earth System Numerical Modeling and Application, Institute of Atmospheric Physics, Chinese Academy of Sciences, Beijing, China. ⁹Department of Climate and Energy Systems Engineering, Ewha Womans University, Seoul, South Korea. ¹⁰Institute for Environmental Studies, Vrije Universiteit Amsterdam, Amsterdam, The Netherlands. ¹¹State Key Laboratory of Climate System Prediction and Risk Management, Nanjing University of Information Science and Technology, Nanjing, China. ¹²Key Laboratory of Meteorological Disaster, Ministry of Education, Nanjing University of Information Science and Technology, Nanjing, China. ¹³Collaborative Innovation Center on Forecast and Evaluation of Meteorological Disasters, Nanjing University of Information Science and Technology, Nanjing, China. ¹⁴Institute for Carbon Neutrality, Tsinghua University, Beijing, China. ✉e-mail: deliangchen@tsinghua.edu.cn

