## [Transparent Peer Review file · Nature Communications]

Human-Induced Westerly Jet Shifts Coordinate Terrestrial Productivity at the Hemispheric Scale

Corresponding Author: Professor Deliang Chen

Version 0:

Reviewer comments:

Reviewer #1

(Remarks to the Author)

Yang et al. identified a shift in the curvature of Northern Hemisphere westerlies around the year 2000 and investigated its influence on GPP. They further examined the contribution of anthropogenic greenhouse gas emissions to this influence. The topic is timely and interesting, as it extends the feedback between hydrometeorological factors and the vegetation carbon cycle to the upstream drivers of large-scale atmospheric circulation. Nonetheless, I have some concerns that need to be clarified before recommending its publication.

Major concerns

- a) The structure of the Results section could be further improved. Currently, it contains only two subsections. The first subsection, "Linking Observed Changes in Westerly Curvature to GPP Responses", is relatively broad in scope and could be divided into two or three clearer components. For example, (1) changes in westerly curvature; (2) influences of westerly curvature changes on GPP; and (3) the underlying mechanisms of these influences.
- b) If possible, the authors may consider incorporating one or more additional GPP products. Several GPP datasets are available for the study period (e.g., EC-LUE, FLUXCOM GPP). While three meteorological reanalysis datasets were analyzed, only one GPP dataset (NIRv) was used in the study.

Minor concerns

Lines 56-68: There is a highly relevant reference regarding the relative importance of temperature and water available on vegetation carbon uptake in the Northern Hemisphere midlatitudes: Wu et al. (2025, GCB, <https://doi.org/10.1111/gcb.70032>). Interestingly, Wu et al. found that the temperature control on vegetation uptake in the NH midlatitudes shifted around the year 2000, the same as the shifting year of westerly curvature change in the present study.

Lines 71, 84, 86: references missing for these statements.

Line 93: It would be clearer to specify the study objective more precisely. The term "carbon cycle" could be replaced with "carbon uptake" or "GPP," as the analysis focuses specifically on GPP responses.

Line 95: 'Results' should be changed to 'Results and discussion' here.

Line 103: I recommend adding one or two sentences to briefly explain what "westerly curvature" represents and how it is calculated. Many readers may not be familiar with this technical term.

Line 115, Fig. 1: Similar to the above comment, it's better to note what absolute, positive, and negative values of curvature mean. If readers have read the methods section, they might understand these specialized terms, but when they start reading the main text from the beginning, they may get lost. The latitude ranges for the statistics in Fig. 1 should be indicated in the caption.

Figure 2: The y-axis label of panel a is missing; the y-axis label 'Slope' of panel b seems to be inconsistent with the description of 'Linear regression coefficients' in the caption (lines 148-149).

Figure 3: "CESM-LENS" is shown in the figure, whereas "CESM1-LENS" is used in the caption and Methods. Please maintain consistent terminology.

Lines 178-186: It is unclear whether these results are derived from Figure 2d. Please clarify.

Line 303: References missing.

Line 330: Variable definitions are missing and should be clearly provided.

Reviewer #2

(Remarks to the Author)

The authors link anthropogenically driven shifts in Northern Hemisphere westerly jet curvature to spatial patterns in terrestrial gross primary productivity (GPP). I appreciate the authors' work on linking land surface processes with atmospheric circulation. I suggest the following addition or expansion to strengthen the mechanistic context and situate the findings in the existing jet dynamics literature:

1. Line 82-93: This paragraph sets up the main objective of the manuscript by arguing that previous studies primarily attribute changes in terrestrial GPP to local climate variability, while the role of large-scale atmospheric circulation remains insufficiently explored. However, the distinction between local hydroclimate drivers and westerly jet dynamics is not clearly established in the manuscript. For example, Cheng et al. (2025) showed that westerly jet waviness modulates mid-latitude hydroclimate variability, particularly precipitation patterns. Therefore, changes in GPP attributed to jet shifts may in fact be mediated through hydroclimate variability rather than representing an independent circulation-driven mechanism.

2. At present, the manuscript does not provide evidence separating the influence of westerly jet curvature from that of local hydroclimate drivers. Without quantifying how much GPP variability is explained by jet shifts independent of precipitation, temperature, and soil moisture, it is difficult to assess whether this study identifies a distinct large-scale control on GPP or is largely reframing already documented hydroclimate effects. It would significantly strengthen the manuscript if the authors could explicitly partition the variance in GPP attributable to (1) local hydroclimate variability and (2) westerly jet shifts and demonstrate the added explanatory power of circulation metrics beyond local drivers. Although spatial correspondence between westerly curvature and GPP is presented, the current analysis relies mainly on regression and correlation, which do not establish causality. The authors are encouraged to consider applying causal inference approaches—such as Granger causality or related methods—to more rigorously assess directional influence and underlying mechanisms. Otherwise, the current framing risks overstating the novelty of the circulation-based perspective.

3. Line 36-37: “The observed spatial pattern of GPP trends closely” The manuscript relies exclusively on the NIRv-based GPP product (1982–2018) to diagnose spatial trends and link them to the reported ~2000 reversal in Northern Hemisphere westerly jet curvature. Bai et al. (2023) demonstrate that GLASS, LRF, NIRv, MODIS, and VPM products show consistent increasing trends before 2000 but diverge significantly thereafter, with regional inconsistencies. I recommend conducting a multi-product sensitivity analysis to evaluate whether the curvature–GPP relationship persists across different GPP products. It would be valuable to discuss how product-specific biases might interact with jet-modulated GPP spatial variability. Without such sensitivity testing, it is unclear whether the reported hemispheric coordination of GPP patterns reflects a dynamically robust signal or is partially dependent on the characteristics of the NIRv product.

4. Figure 2 highlights regions (specifically R2, R5, and also Southeast Asia) that overlap substantially with cropland-dominated areas. These regions are agricultural managed systems, where human influence, such as irrigation, fertilisation, crop rotation, and land-use practices exert strong control over GPP variability. In such landscapes, anthropogenic management can significantly modulate or even override climate-driven constraints on productivity. While large-scale westerly jet curvature shifts may influence regional hydroclimate (e.g., precipitation patterns, temperature extremes, or radiation anomalies), their direct control over GPP in managed croplands may be weaker or strongly mediated by human interventions. For example, irrigation can buffer precipitation deficits, and fertilizer inputs can enhance photosynthetic capacity independent of circulation-induced variability. Given that the manuscript emphasizes hemispheric-scale coordination of GPP patterns driven by jet curvature changes, it would be important to clarify whether the reported relationships hold consistently across different land-cover types, particularly between natural ecosystems and intensively managed croplands.

5. Line 189-225: While the CESM experiments suggest GHG dominance over aerosols and biomass burning in the modelled response of Northern Hemisphere westerly curvature changes, the physical pathway linking GHG forcing to these curvature changes requires clearer articulation. Explicitly connecting the diagnosed curvature changes to established dynamical mechanisms would strengthen confidence in the attribution and clarify how this relates to existing westerly jet-shift drives literature, e.g. Arctic warming (Cheng et al 2025), tropical warming (Woollings et al 2023).

References

Cheng, L., Zhang, J., Wu, Y. et al. Westerly jet waviness modulates mid-latitude hydroclimate variability. *Nat Commun* 16, 10928 (2025). <https://doi.org/10.1038/s41467-025-65904-8>

Bai, Y., Liang, S., Jia, A., & Li, S. (2023). Different satellite products revealing variable trends in global gross primary production. *Journal of Geophysical Research: Biogeosciences*, 128(7), e2022JG006918.

Woollings, T., Drouard, M., O'Reilly, C.H. et al. Trends in the atmospheric jet streams are emerging in observations and could be linked to tropical warming. *Commun Earth Environ* 4, 125 (2023). <https://doi.org/10.1038/s43247-023-00792-8>

Version 1:

Reviewer comments:

Reviewer #1

(Remarks to the Author)

(Remarks on code availability)

Reviewer #2

(Remarks to the Author)

The authors have done a significant amount of work to improve the manuscript, and it now looks much better. I have a few minor comments:

1. While the authors attempt to decompose the influence of westerly curvature into “direct” and “indirect” components, the physical interpretation of these pathways remains somewhat unclear. In Text S1, “The information flow from westerly curvature to GPP is 0.26, indicating that westerly curvature is an important driver and direct influencing factor of GPP variability.” The direct pathway (westerly curvature → GPP) assumes the existence of a mechanism independent of hydroclimate variables, which is not fully explained. It may be helpful to briefly discuss the physical plausibility of such a direct pathway independent of precipitation and temperature, or to adopt more cautious wording, rather than implying a strictly independent causal mechanism, especially when the manuscript acknowledges the difficulty of isolating the influence of individual variables in a coupled earth system.
2. Does the Liang–Kleeman information flow approach applied here require the underlying time series to be stationary, particularly given the apparent reversal in westerly jet curvature around the year 2000?
3. Like other causal discovery approaches, the Liang–Kleeman Information Flow also depends on sample size. It would be helpful to report the sample size, as in data-driven approaches with small datasets, we cannot confidently conclude that the inferred causal network represents true causal relationships.
4. The sign of the relationship is consistent across different GPP datasets; however, the strength and the number of statistically significant pixels vary, despite all datasets covering the same period. This is particularly evident for the LUE-based GPP product, which shows relatively few statistically significant pixels. This difference could be more clearly illustrated by identifying or quantifying consistent and inconsistent pixels across datasets. A brief comment (1–2 lines) on uncertainties associated with different GPP products would also be helpful.
5. Lines 68–69: The cited reference uses NDVI, and NDVI and GPP may diverge. It would be useful to explicitly mention the variable used in the citation to avoid confusion. For example:
“Recent land vegetation growth, observed through satellite-derived Normalised Difference Vegetation Index (NDVI), is regulated by natural climate variability such as the El Niño–Southern Oscillation (ENSO) and anthropogenic climate change.”
6. Figure 1 caption currently explains only panels (a) and (b). It would be helpful to include brief descriptions of panels (c–k) to improve clarity for the reader.
7. Line 225: Consider replacing “ecosystem productivity” with “vegetation productivity” or simply “GPP” for clarity.
8. Lines 284–286: The statement that “the difference between GPP trends under greenhouse gas and non-greenhouse gas forcing closely matches the observed spatial pattern” may need reconsideration. In Western North America, observations indicate a declining GPP trend, whereas the GHG-minus-non-GHG difference suggests an increase, indicating a mismatch in this region.

(Remarks on code availability)

Version 2:

Reviewer comments:

Reviewer #2

(Remarks to the Author)

The authors have addressed all of my comments satisfactorily. I recommend acceptance.

(Remarks on code availability)

General Response to the Reviewers' Comments

Dear Editor and Referees,

We would like to express our deepest gratitude to the three referees for their exceptionally rigorous, and constructive feedback. Your professional insights and the significant time you invested in reviewing our manuscript have been instrumental in refining our work.

The key improvements in this revision, following your guidance, are summarized below:

- Incorporated 3 additional GPP datasets to validate our results and ensure the robustness of our conclusions.
- Added a causal network analysis linking westerly jet curvature, circulation factors, surface (hydro-) climate variables, and GPP to further examine the underlying mechanisms.
- Assessed how land-use type modulates the effect of westerly jet curvature on GPP.
- Conducted a more comprehensive literature review and expanded the discussion to contextualize our findings.
- To accommodate the additional analyses and figures, we have included Supplementary Information Text S1, which provides detailed causal analyses of westerly jet curvature regulating GPP. Furthermore, in accordance with the journal's requirement that the number of extended figures does not exceed 10, the numbering of relatively important supplementary figures has been adjusted to Extended Figure 1–10, while the numbering of relatively minor figures, such as those for multi-source data validation, has been adjusted to Figure S1–S6.
- The manuscript was thoroughly reviewed and revised, with unclear or omitted statements clarified.

The above points summarize the major revisions we have made. We have also tracked and highlighted the major revisions made in response to each comment in the revised manuscript. For further details, please refer to our point-by-point responses. The reviewers' comments are shown in **black**, our responses are provided in **blue**, and the corresponding changes in the manuscript are highlighted in **red**.

Response Letter

REVIEWERS' COMMENTS:

Reviewer #1

Overall comment: Yang et al. identified a shift in the curvature of Northern Hemisphere westerlies around the year 2000 and investigated its influence on GPP. They further examined the contribution of anthropogenic greenhouse gas emissions to this influence. The topic is timely and interesting, as it extends the feedback between hydrometeorological factors and the vegetation carbon cycle to the upstream drivers of large-scale atmospheric circulation. Nonetheless, I have some concerns that need to be clarified before recommending its publication.

Response: We sincerely thank the reviewer for recognizing the significance of our work. Your valuable suggestions have helped us improve the logical flow and structure of the manuscript, and also refine many details, making our conclusions more robust. Below, we provide our detailed responses to your comments.

Major concerns

1. The structure of the Results section could be further improved. Currently, it contains only two subsections. The first subsection, “Linking Observed Changes in Westerly Curvature to GPP Responses”, is relatively broad in scope and could be divided into two or three clearer components. For example, (1) changes in westerly curvature; (2) influences of westerly curvature changes on GPP; and (3) the underlying mechanisms of these influences.

Response: We appreciate your suggestion, which has helped us optimize the structure of the manuscript. Following your recommendation, the section “Linking Observed Changes in Westerly Curvature to GPP Responses” has been divided into two subsections to maintain a clear structure while avoiding overly brief sections. The original subsection “*Linking Observed Changes in Westerly Curvature to GPP Responses*” has been retained to include Figures 1 (changes in the westerly curvature) and 2 (relationship between the westerly and GPP). In addition, a new subsection, “*Underlying Mechanisms of GPP Responses to Westerly Curvature Changes*”, has been added to cover Figure 3, which illustrates the mechanism by which westerly curvature influences GPP through local circulation.

2. If possible, the authors may consider incorporating one or more additional GPP products. Several GPP datasets are available for the study period (e.g., EC-LUE, FLUXCOM GPP). While three meteorological reanalysis datasets were analyzed, only one GPP dataset (NIRv) was used in the study.

Response Letter

Response: We appreciate the reviewer's insightful comment, which has helped enhance the robustness of our findings. In response, we have included three additional GPP datasets to provide supplementary validation:

(1). The GLASS project provides GPP (GLASS GPP) products with a long period of coverage (1982–2018) (Liang et al., 2021). The algorithm of this GPP product was proposed by Yuan et al., using the eddy covariance-light use efficiency (EC-LUE) model, which integrates 8 LUE models used extensively worldwide (Yuan et al., 2007, 2010).

(2). Global monthly GPP from an improved LUE Model during 1982-2016 (Madani et al., 2017). This dataset (LUE GPP) was improved with optimized spatially and temporally explicit LUE values derived from selected FLUXNET tower site data. Global gridded long-term daily GPP was derived using the optimized LUE, Global Inventory Modeling and Mapping Studies (GIMMS3g) canopy fraction of photosynthetically active radiation (FPAR), and Modern-Era Retrospective analysis for Research and Applications, Version 2, (MERRA-2) meteorological information. This dataset provides satellite-based GPP estimates derived from a refined LUE-model framework.

(3). LRF GPP, a GPP product proposed by Tagesson et al. (2021). It is a new method to estimate the ecosystem-level physiological approach of GPP using the asymptotic LRF between GPP and incoming photosynthetically active radiation. This GPP dataset ranges from 1982 to 2016.

We specifically selected these datasets because they provide longer temporal coverage and spatially gridded information. Most other GPP products only cover a relatively short period after 2000, whereas the curvature of the Northern Hemisphere westerly jet, which is central to our study, changed around 2000. Using short-term post-2000 data alone would not fully capture the relationship between GPP and jet curvature. Additionally, although FLUXNET observations and FLUXCOM dataset are invaluable, their spatial coverage is sparse in many of our hotspot regions, and the record lengths are limited. The three datasets we selected include LUE GPP, which extensively utilized FLUXNET data for gap-filling and optimization, while GLASS GPP and LRF GPP have also been thoroughly validated in previous studies, demonstrating good agreement with observational data and strong applicability.

Our analysis shows that while absolute values and trends differ across GPP products due to variations in retrieval and calculation methods, the overall spatial pattern of GPP associated with westerly jet curvature in our hotspot regions (R1–R5) remains unchanged. This indicates that the influence of westerly jet curvature on the large-scale spatial GPP patterns in mid- to high-latitude Northern Hemisphere hotspots is a robust mechanism, independent of the specific GPP dataset used. In regions outside the hotspots considered in this study, the responses of different GPP products to westerly jet curvature exhibit slight spatial differences (e.g., in eastern China and tropical Africa), but the overall pattern remains highly consistent, with spatial correlation coefficients between products that are significant at the 95% confidence level.

Response Letter

We have added the following description in the "Reanalysis and Observation Datasets" section (Lines 357 to 367): To evaluate whether the robustness of GPP responses to westerly jet curvature is affected by dataset differences, we additionally used three GPP products for validation (the additional GPP data are used Figure S4): (1) GLASS GPP, covering 1982–2018 at 0.05° resolution, derived using an eddy covariance–light use efficiency (EC-LUE) model that integrates multiple globally used LUE models (Liang et al., 2021; Yuan et al., 2007, 2010). (2) LUE GPP, a global monthly GPP product covering 1982–2016 at 8 km resolution, estimated with an optimized LUE model based on FLUXNET data, combined with GIMMS3g FPAR and MERRA-2 meteorology data (Madani et al., 2017). (3) LRF GPP, spanning 1982–2016 at 0.05° resolution, estimated using an asymptotic light response function between GPP and incoming photosynthetically active radiation data (Tagesson et al., 2021). The results show that while minor spatial differences exist among datasets (e.g., in tropical Africa and eastern China), the overall spatial pattern in mid- to high-latitude regions is highly consistent, particularly within the hotspots considered in this study.

Response Letter

Figure S4. Relationships between summer westerly patterns in the Northern Hemisphere and gross primary productivity (GPP) based on different products. Spatial distributions of correlation coefficients between local summer GPP and Northern Hemisphere overall curvature (1982–2018) using (a) GLASS GPP, (b) LUE GPP, and (c) LRF GPP products. Only correlations significant at the 95% confidence level are shown.

References

Liang, S. et al. The Global Land Surface Satellite (GLASS) Product Suite. *Bulletin of the American Meteorological Society* 102, E323–E337 (2021).

Madani, N., Kimball, J. S. & Running, S. W. Improving Global Gross Primary Productivity Estimates by Computing Optimum Light Use Efficiencies Using Flux Tower Data. *JGR Biogeosciences* 122, 2939–2951 (2017).

Tagesson, T. et al. A physiology-based Earth observation model indicates stagnation in the global gross primary production during recent decades. *Global Change Biology* 27, 836–854 (2021).

Yuan, W. et al. Global estimates of evapotranspiration and gross primary production based on MODIS and global meteorology data. *Remote Sensing of Environment* 114, 1416–1431 (2010).

Yuan, W. et al. Deriving a light use efficiency model from eddy covariance flux data for predicting daily gross primary production across biomes. *Agricultural and Forest Meteorology* 143, 189–207 (2007).

Minor concerns

1. Lines 56-68: There is a highly relevant reference regarding the relative importance of temperature and water available on vegetation carbon uptake in the Northern Hemisphere midlatitudes: Wu et al. (2025, *GCB*, <https://doi.org/10.1111/gcb.70032>). Interestingly, Wu et al. found that the temperature control on vegetation uptake in the NH midlatitudes shifted around the year 2000, the same as the shifting year of westerly curvature change in the present study.

Response: We thank you for providing this valuable information. We have appropriately cited the references you provided, as well as related literature, in both the *Introduction*, and *Conclusion and Discussion* sections. We have added citations to the following references:

Piao S, Liu Z, Wang T, et al. Weakening temperature control on the interannual variations of spring carbon uptake across northern lands[J]. *Nature Climate Change*, 2017, 7(5): 359-363.

Response Letter

Wu H, Fu C, Yu K, et al. Drought-Induced Weakening of Temperature Control on Ecosystem Carbon Uptake Across Northern Lands[J]. *Global Change Biology*, 2025, 31(1): e70032.

Wu H, Fu C, Zhang L, et al. Significant sensitivity of global vegetation productivity to terrestrial surface wind speed changes[J]. *Nature Communications*, 2025, 16(1): 9315.

Zhu P, Zhuang Q, Welp L, et al. Recent warming has resulted in smaller gains in net carbon uptake in northern high latitudes[J]. *Journal of Climate*, 2019, 32(18): 5849-5863.

We have also added the following descriptions in the Results section (Lines 197 to 200): **A recent study also indirectly supports our findings from a different perspective (Wu et al., 2025), reporting that around the year 2000, the temperature control on vegetation uptake of carbon in the Northern Hemisphere midlatitudes underwent a shift, consistent with the timing of the westerly curvature changes identified in the present study.**

2. Lines 71, 84, 86: references missing for these statements.

Response: We thank the reviewer for this helpful reminder. We have added appropriate references to support these statements.

References for Line71:

Beer, C. et al. Terrestrial Gross Carbon Dioxide Uptake: Global Distribution and Covariation with Climate. *Science* 329, 834–838 (2010).

Gampe, D. et al. Increasing impact of warm droughts on northern ecosystem productivity over recent decades. *Nat. Clim. Chang.* 11, 772–779 (2021).

Liu, Z. et al. Precipitation shapes the spatial pattern of gross primary productivity, while temperature drives its interannual variability in the Northern Hemisphere. *Forest Ecology and Management* 601, 123328 (2026).

Xia, J. et al. Joint control of terrestrial gross primary productivity by plant phenology and physiology. *Proc. Natl. Acad. Sci. U.S.A.* 112, 2788–2793 (2015).

References for Line84:

Chen, S., Xiao, J., Li, X., Wu, M. & Yang, J. Disentangling the climate–VPD–GPP Nexus: Global patterns and underlying drivers. *Global and Planetary Change* 256, 105141 (2026).

Fang, Z. et al. Global increase in the optimal temperature for the productivity of terrestrial ecosystems. *Communications Earth & Environment* 5, 466 (2024).

Response Letter

Liu, Z. et al. Precipitation thresholds regulate net carbon exchange at the continental scale. *Nature Communications* 9, 3596 (2018).

Wang, Y., Li, P., Yuemin, Y. & Tiantian, C. Global Vegetation-Temperature Sensitivity and Its Driving Forces in the 21st Century. *Earth's Future* 12, e2022EF003395 (2024).

References for Line86:

Kornhuber, K. et al. Extreme weather events in early summer 2018 connected by a recurrent hemispheric wave-7 pattern. *Environ. Res. Lett.* 14, 054002 (2019).

Kornhuber, K., Petoukhov, V., Petri, S., Rahmstorf, S. & Coumou, D. Evidence for wave resonance as a key mechanism for generating high-amplitude quasi-stationary waves in boreal summer. *Climate Dynamics* 49, 1961–1979 (2016).

Lian, X., Li, Y., Liu, J., Kornhuber, K. & Gentine, P. Northern ecosystem productivity reduced by Rossby-wave-driven hot-dry conditions. *Nature Geoscience* 18, 615–623 (2025).

Petoukhov, V. et al. Role of quasiresonant planetary wave dynamics in recent boreal spring-to-autumn extreme events. *Proceedings of the National Academy of Sciences* 113, 6862–6867 (2016).

Yang, X. et al. Phase-Locked Rossby Wave-4 Pattern Dominates the 2022-Like Concurrent Heat Extremes Across the Northern Hemisphere. *Geophysical Research Letters* 51, e2023GL107106 (2024).

3. Line 93: It would be clearer to specify the study objective more precisely. The term “carbon cycle” could be replaced with “carbon uptake” or “GPP,” as the analysis focuses specifically on GPP responses.

Response: We thank the reviewer for the suggestion. It has been changed to “**carbon uptake**”.

4. Line 95: ‘Results’ should be changed to ‘Results and discussion’ here.

Response: We thank the reviewer for the suggestion. Considering that this section primarily presents the results, with only minimal discussion provided to aid interpretation and support the findings, we have decided not to add discussion here. However, we have revised the title of the Conclusions section to “**Conclusions and Discussion**”, as this section covers our main interpretations and perspectives.

5. Line 103: I recommend adding one or two sentences to briefly explain what “westerly curvature” represents and how it is calculated. Many readers may not be familiar with this technical term.

Response Letter

Response: Thanks for your suggestion. We have added a brief description after the first mention of ‘curvature’ (Lines 112 to 117): Briefly, we computed the mathematical curvature at each point along the westerly jet axis at 200 hPa using ERA5 data (referred to as westerly curvature), which is defined by a series of longitude–latitude coordinates. It provides a quantitative measure of the local streamline geometry, with positive curvature indicating a northward ridge, and negative curvature a southward trough (see the “Indicator for westerlies curvature” section in Data and Method for methodological details).

6. Line 115, Fig. 1: Similar to the above comment, it’s better to note what absolute, positive, and negative values of curvature mean. If readers have read the methods section, they might understand these specialized terms, but when they start reading the main text from the beginning, they may get lost. The latitude ranges for the statistics in Fig. 1 should be indicated in the caption.

Response: Thank you for pointing out the readability issue. While addressing your Minor concern 5, we have added a brief explanation of the curvature signs: It provides a quantitative measure of the local streamline geometry, with positive curvature indicating a northward ridge, and negative curvature a southward trough (see the “Indicator for westerlies curvature” section in Data and Method for methodological details). (Lines 112 to 117).

In response to your second concern, we have added the following information to the caption of Figure 1 to aid reader understanding (Lines 150 to 152): The basemaps in (a) and (b) are shown only for reference of the longitude range on the x-axis. The y-axis represents curvature and is unrelated to latitude. The westerly jet axis detection is performed within the Northern Hemisphere (0–90° N). We have also added the detection range of the westerly axis in the ‘Detection of Westerlies Changes in the Northern Hemisphere’ section (Lines 419 to 420): We first applied morphological operations to daily 200 hPa zonal wind fields over the Northern Hemisphere (0–90° N).

7. Figure 2: The y-axis label of panel a is missing; the y-axis label ‘Slope’ of panel b seems to be inconsistent with the description of ‘Linear regression coefficients’ in the caption (lines 148-149).

Response: We have updated Figure 2 by adding the y-axis label to panel a. For panel b, the y-axis label ‘Slope’ refers to the regression slope; to avoid misunderstanding, we have changed it to ‘Regression Coefficient’. Since this is a regression of local curvature onto global curvature, the regression coefficient is unitless.

Response Letter

Figure 2. Relationships between summer westerly patterns in the Northern Hemisphere and gross primary productivity (GPP). (a) Overall curvature anomaly (see Eq. 3 in Data and Method, 10^{-5} degree) of summer westerlies in the Northern Hemisphere (1979–2023). Dashed lines indicate ± 0.5 standard deviation. (b) Linear regression coefficients (unitless) of local westerly curvature onto overall curvature along each longitude. Red-shaded areas are significant at the 95% confidence level. (c) Spatial distribution of the correlation coefficients between local summer GPP and Northern Hemisphere overall curvature during 1982–2018. Only correlations significant at the 95% confidence level are shown. (d) Trend of GPP ($\text{g C m}^{-2} \text{ yr}^{-1}$) in the Northern Hemisphere during 1982–2018. Only statistically significant trends at the 95% confidence level are shown.

8. Figure 3: “CESM-LENS” is shown in the figure, whereas “CESM1-LENS” is used in the caption and Methods. Please maintain consistent terminology.

Response: Thank you for pointing this out. We have now standardized all related descriptions to ‘CESM1-LENS’.

9. Lines 178-186: It is unclear whether these results are derived from Figure 2d. Please clarify.

Response Letter

Response: We apologize for the missing figure references. These conclusions are mainly based on Figures 2c and 2d, as well as Extended Figures 5 and 6. We have carefully reviewed the entire manuscript and ensured more comprehensive figure referencing throughout.

10. Line 303: References missing.

Response: We have added the relevant references, which used and evaluated this dataset, confirming its applicability and utility.

Badgley, G., Field, C. B. & Berry, J. A. Canopy near-infrared reflectance and terrestrial photosynthesis. *Sci. Adv.* 3, e1602244 (2017).

Chen, Z., Wang, W., Forzieri, G. & Cescatti, A. Transition from positive to negative indirect CO₂ effects on the vegetation carbon uptake. *Nat Commun* 15, 1500 (2024).

Wang, S., Zhang, Y., Ju, W., Qiu, B. & Zhang, Z. Tracking the seasonal and inter-annual variations of global gross primary production during last four decades using satellite near-infrared reflectance data. *Science of The Total Environment* 755, 142569 (2021).

Zhu, W. et al. Remote sensing of terrestrial gross primary productivity: a review of advances in theoretical foundation, key parameters and methods. *GIScience & Remote Sensing* 61, 2318846 (2024).

11. Line 330: Variable definitions are missing and should be clearly provided.

Response: We apologize for the lack of clarity. We have now provided complete definitions for all variables in Equation 1 (Lines 399 to 404):

$$\begin{cases} \Delta C_{GHG} = \frac{(C_{ALL} - C_{XGHG})}{C_{ALL}} \times 100\% \\ \Delta C_{AER} = \frac{(C_{ALL} - C_{XAER})}{C_{ALL}} \times 100\% \\ \Delta C_{BMB} = \frac{(C_{ALL} - C_{XBMB})}{C_{ALL}} \times 100\% \end{cases} \quad (1)$$

In Equation (1), ΔC_{GHG} , ΔC_{AER} , and ΔC_{BMB} represent the contribution of greenhouse gas emissions, aerosol emissions, and biomass burning emissions, respectively, to changes in variable C (e.g., surface air temperature, precipitation, GPP). C_{ALL} refers to variable C in the all-forcing experiment, while C_{XGHG} , C_{XAER} , and C_{XBMB} denote variable C in the fixed greenhouse gas, aerosol, and biomass burning forcing experiment, respectively.

Response Letter

Reviewer #2

Overall comment: The authors link anthropogenically driven shifts in Northern Hemisphere westerly jet curvature to spatial patterns in terrestrial gross primary productivity (GPP). I appreciate the authors' work on linking land surface processes with atmospheric circulation. I suggest the following addition or expansion to strengthen the mechanistic context and situate the findings in the existing jet dynamics literature:

References

Cheng, L., Zhang, J., Wu, Y. et al. Westerly jet waviness modulates mid-latitude hydroclimate variability. *Nat Commun* 16, 10928 (2025). <https://doi.org/10.1038/s41467-025-65904-8>
Bai, Y., Liang, S., Jia, A., & Li, S. (2023). Different satellite products revealing variable trends in global gross primary production. *Journal of Geophysical Research: Biogeosciences*, 128(7), e2022JG006918.
Woollings, T., Drouard, M., O'Reilly, C.H. et al. Trends in the atmospheric jet streams are emerging in observations and could be linked to tropical warming. *Commun Earth Environ* 4, 125 (2023). <https://doi.org/10.1038/s43247-023-00792-8>

Response: Thank you for your careful review and for recognizing the value of our research approach. We sincerely appreciate the time and effort you devoted to evaluating our manuscript. In response to your comments, we have carefully revised the manuscript. The references you suggested are highly valuable and closely related to our study, and they have helped us supplement important information in the relevant sections. We have incorporated and cited these references at appropriate places in the revised manuscript (Lines 54, 74 to 77, and 79 to 81).

Other Comment:

1. Line 82-93: This paragraph sets up the main objective of the manuscript by arguing that previous studies primarily attribute changes in terrestrial GPP to local climate variability, while the role of large-scale atmospheric circulation remains insufficiently explored. However, the distinction between local hydroclimate drivers and westerly jet dynamics is not clearly established in the manuscript. For example, Cheng et al. (2025) showed that westerly jet waviness modulates mid-latitude hydroclimate variability, particularly precipitation patterns. Therefore, changes in GPP attributed to jet shifts may in fact be mediated through hydroclimate variability rather than representing an independent circulation-driven mechanism.

Response: Thank you for this insightful comment. We agree that changes in the westerly jet can influence terrestrial GPP through their modulation of regional hydroclimate, as highlighted in studies such as Cheng et al. (2025). Combined with your second comment, we applied the Liang–Kleeman information flow method to construct a causal pathway linking westerly curvature changes, local

Response Letter

(hydroclimate) climate factors, and GPP. Compared with the traditional Granger causality approach, the information flow approach provides a more physically interpretable measure of causality, as it is derived from dynamical system theory and quantifies the directional transfer of information between variables. Unlike Granger causality, which is primarily based on predictive capability within linear statistical models, the information flow is better suited for analysing complex and nonlinear interactions in climate systems. This approach allows us to partially disentangle the direct influence of westerly variability from indirect effects mediated through hydroclimate and other climate factors. Considering the constraints on main-text length and the prioritization of key findings, the detailed methodology of the information flow analysis is described in the Supporting Information (Text S1):

Text S1. Causal pathways of westerly curvature regulating gross primary productivity in the mid-to-high latitudes of the Northern Hemisphere

Within our framework, we emphasize treating large-scale westerly circulation changes as an organizing dynamical mode that can restructure regional climate conditions and thereby regulate spatial patterns of GPP, while also attempting, to some extent, to explicitly identify the direct dynamical effects of westerly jet variability from indirect pathways mediated through hydroclimate or land–atmosphere interactions.

Here we introduce Liang-Kleeman Information Flow Theory (Liang, 2014; Rong and Liang, 2021). Compared with the traditional Granger causality approach, the information flow approach provides a more physically interpretable measure of causality, as it is derived from dynamical system theory and quantifies the directional transfer of information between variables. Unlike Granger causality, which is primarily based on predictive capability within linear statistical models, the information flow approach is better suited for analysing complex and nonlinear interactions in climate systems (Stips et al., 2016; Sun et al., 2025). Using this theory, the causal relationship can be determined only by the time series of the two parameters. According to the theory, under the assumption of linear mode, the maximum likelihood estimation form of the information flow from X_2 to X_1 (X_1 and X_2 are two time series) is:

$$T_{2 \rightarrow 1} = \frac{C_{11}C_{12}C_{2,d1} - C_{12}^2C_{1,d1}}{C_{11}^2C_{22} - C_{11}C_{12}^2}$$

where C_{ij} is the covariance of X_i and X_j and $C_{i,dj}$ is the covariance of X_i and $\{(X_{j,n+1} - X_{j,n})/\delta t\}$, with δt represents the time interval. According to the theory, causation implies correlation. However, the converse is that correlation does not imply causation. When $|T_{2 \rightarrow 1}| > 0$, X_2 is a cause of X_1 ; and when $|T_{2 \rightarrow 1}| = 0$, X_2 is not the cause of X_1 . $|T_{2 \rightarrow 1}|$ represents the information flow strength from X_2 to X_1 , with larger values indicating stronger causal associations. To assess causal strength in a multivariate system, we used standardized information flow.

In causal network analysis, the indirect effect transmitted along a causal chain is typically calculated as the product of the path coefficients along that chain (Mackinnon et al., 2007; Hair et al., 2021).

Response Letter

Therefore, the indirect contribution of $A \rightarrow C$ mediated through B can be approximated as $T_{A \rightarrow B} \cdot T_{B \rightarrow C}$. Based on these, the information flow from A to C is decomposed into direct influence, along the direct $A \rightarrow C$ path, and indirect influence, along all paths from A to C that pass through other nodes. LK information flow can be computed using the Python package LK-Info-Flow (<https://pypi.org/project/LK-Info-Flow/>).

Using the information flow approach, we computed standardized information flow matrices (Extended Figure 7a) for five key sectors, including Eastern Europe, Central Asia, East Asia, Western North America, and Central North America, covering westerly curvature, circulation factors (200 hPa geopotential height, zonal and meridional winds), local climate variables (surface maximum temperature and precipitation), and GPP. From these matrices, the main causal network (Extended Figure 7b) structures were extracted (Supporting Information Text S1):

We applied the information flow approach to quantify the mutual information flow strength among westerly curvature, circulation factors (200 hPa geopotential height, zonal and meridional winds), local climate variables (surface maximum temperature and precipitation), and GPP within five key sectors: Eastern Europe, Central Asia, East Asia, Western North America, and Central North America. Based on the information flow matrices, we constructed a critical causal network to further identify the pathways through which westerly curvature influences terrestrial GPP. Specifically, we extracted causal connections with standardized information flow strength greater than 0.1, corresponding to approximately the top 60% of all possible connections, to highlight the most significant causal pathways.

Westerly curvature primarily reflects the bending of the westerly jet rather than its intensity, which explains why bidirectional causal links between westerly curvature, geopotential height, and meridional wind are stronger, while causal connections with zonal wind are relatively weak. The information flow from westerly curvature to GPP is 0.26, indicating that westerly curvature is an important driver and direct influencing factor of GPP variability. Geopotential height and meridional wind also contribute to changes in surface maximum temperature and precipitation, thereby directly or indirectly regulating GPP. These results suggest that variations in westerly curvature can directly affect GPP through dynamical adjustments of circulation patterns and indirectly influence GPP by modulating land-surface (hydro-) climate factors such as temperature and precipitation. The key causal network also clearly reflects the strong coupling and complexity of these systems. Under such circumstances, it is impossible to completely isolate the contribution of any single factor or process. Therefore, we adopted an idealized estimation approach: the direct information flow from westerly curvature to GPP is regarded as the magnitude of the direct effect, while the sum of all remaining pathway information flows from westerly curvature to GPP is considered the indirect effect of westerly curvature on GPP. The indirect effect is calculated as the sum of the information flows along all pathways from the westerly curvature to GPP that ultimately reach GPP through the nodes of maximum surface air temperature or precipitation. For

Response Letter

each pathway, the information flow is computed as the product of the information flows of its individual segments (Mackinnon et al., 2007; Hair et al., 2021). The idealized estimation results indicate that the direct effect (0.26) accounts for approximately 57.8% of the total influence and is about 1.4 times larger than the indirect effect (0.19). It should also be noted that the total effect of the westerly curvature (the sum of the direct and indirect effects) remains smaller than the combined direct effects of maximum surface air temperature and precipitation on GPP. This relationship persists even when additional surface (hydro-) climate variables are introduced into the causal network, highlighting the dominant role of surface hydroclimatic conditions in regulating GPP variability. In other words, although variations in westerly curvature can influence ecosystem productivity through large-scale dynamical adjustments (e.g., through curvature-induced wind speed changes (Kidston et al., 2015; Wu et al., 2025)), their impacts are ultimately manifested and amplified through local temperature and precipitation conditions that exert more immediate control on GPP. Another noteworthy aspect is the distinctly central role of geopotential height anomalies within the system. This is evidenced not only by their stronger causal linkages with other factors, but also by their pronounced direct influence on GPP. Such a pattern indicates that geopotential height serves as an effective proxy for atmospheric circulation dynamics, exerting influence on GPP both indirectly, through the regulation of other variables, and directly via dynamical processes. As with most data-driven causal analysis methods, the inferred links reflect predictive associations and uncertainties rather than fully established physical connections operating in the real world, and should therefore be interpreted with caution. In particular, the results should not be taken to imply an artificial separation between circulation dynamics and other influencing factors, since these processes are inherently and fully coupled.

In the section *Underlying Mechanisms of GPP Responses to Westerly Curvature Changes*, we have added the following description (Lines 222 to 230):

In our analysis, large-scale variations in westerly circulation are conceptualized as a primary dynamical driver that structures regional climate and shapes the spatial distribution of GPP. Causal analysis further reveals that, although variations in westerly curvature exert broad-scale influences on ecosystem productivity through circulation adjustments, their effects are ultimately mediated and amplified by local temperature and precipitation, which exert more immediate control over GPP. A detailed description of the causal analysis framework used to identify and quantify the direct and indirect influences of westerly curvature, circulation factors, and surface climate variables on GPP is provided in the Supporting Information (Text S1 and Extended Figure 7).

Response Letter

b. Critical Causal Network

Extended Figure 7. Causal network depicting the influence of westerly curvature on terrestrial gross primary production. (a) Standardized information flow matrix among westerly curvature, circulation factors including 200 hPa geopotential height (Z200), zonal (U200) and meridional (V200) winds, local climate variables including surface maximum temperature (Tmax) and precipitation, and gross primary production (GPP). The color of each circle reflects the mean value across the five sectors (Eastern Europe, Central Asia, East Asia, Western North America, and Central North America), and

Response Letter

the size corresponds to the standard deviation among the sectors. **(b)** The critical causal network, obtained by filtering out information flows below 0.1, represents approximately the top 60% of connections in the fully connected network.

References

Hair, J. F. et al. Mediation Analysis. in Partial Least Squares Structural Equation Modeling (PLS-SEM) Using R 139–153 (Springer International Publishing, Cham, 2021). doi:10.1007/978-3-030-80519-7_7.

Liang, X. S. Unraveling the cause-effect relation between time series. *Phys. Rev. E* 90, 052150 (2014).

MacKinnon, D. P., Fairchild, A. J. & Fritz, M. S. Mediation Analysis. *Annu. Rev. Psychol.* 58, 593–614 (2007).

Rong, Y. & Liang, X. S. Panel Data Causal Inference Using a Rigorous Information Flow Analysis for Homogeneous, Independent and Identically Distributed Datasets. *IEEE Access* 9, 47266–47274 (2021).

Stips, A., Macias, D., Coughlan, C., Garcia-Gorriz, E. & Liang, X. S. On the causal structure between CO₂ and global temperature. *Sci Rep* 6, 21691 (2016).

Sun, J. et al. Causal pathways underlying global soil moisture–precipitation coupling. *Nat Commun* 16, 8935 (2025).

Wu H, Fu C, Zhang L, et al. Significant sensitivity of global vegetation productivity to terrestrial surface wind speed changes[J]. *Nature Communications*, 2025, 16(1): 9315.

Kidston J, Scaife A A, Hardiman S C, et al. Stratospheric influence on tropospheric jet streams, storm tracks and surface weather[J]. *Nature Geoscience*, 2015, 8(6): 433-440.

2. At present, the manuscript does not provide evidence separating the influence of westerly jet curvature from that of local hydroclimate drivers. Without quantifying how much GPP variability is explained by jet shifts independent of precipitation, temperature, and soil moisture, it is difficult to assess whether this study identifies a distinct large-scale control on GPP or is largely reframing already documented hydroclimate effects. It would significantly strengthen the manuscript if the authors could explicitly partition the variance in GPP attributable to (1) local hydroclimate variability and (2) westerly jet shifts and demonstrate the added explanatory power of circulation metrics beyond local drivers. Although spatial correspondence between westerly curvature and GPP is presented, the current analysis relies mainly on regression and correlation, which do not establish causality. The authors are encouraged to consider applying causal inference approaches—such as Granger causality or related methods—to more rigorously assess directional influence and underlying mechanisms. Otherwise, the current framing risks overstating the novelty of the circulation-based perspective.

Response Letter

Response: Thank you for highlighting this point. Causal analysis remains a critical yet challenging aspect of Earth science research, and your comments have been invaluable in improving the completeness of the mechanistic discussion in our manuscript. As this comment overlaps with the first major point, we have provided a comprehensive and cohesive response in our reply to Comment 1 to ensure completeness and clarity. In summary, we applied the Liang-Kleeman information flow framework to quantify the bidirectional causal strengths among westerly curvature, circulation factors, surface (hydro-) climatic variables, and GPP. This approach allowed us to disentangle the direct and indirect effects of westerly curvature on GPP and to identify the role of surface hydroclimatic factors in mediating these influences. These results indicate that, although variations in westerly curvature can affect ecosystem productivity through large-scale dynamical adjustments, their ultimate impacts are realized and amplified via local temperature and precipitation, which exert more immediate control over GPP (Lines 222 to 230).

3. Line 36-37: “The observed spatial pattern of GPP trends closely” The manuscript relies exclusively on the NIRv-based GPP product (1982–2018) to diagnose spatial trends and link them to the reported ~2000 reversal in Northern Hemisphere westerly jet curvature. Bai et al. (2023) demonstrate that GLASS, LRF, NIRv, MODIS, and VPM products show consistent increasing trends before 2000 but diverge significantly thereafter, with regional inconsistencies. I recommend conducting a multi-product sensitivity analysis to evaluate whether the curvature–GPP relationship persists across different GPP products. It would be valuable to discuss how product-specific biases might interact with jet-modulated GPP spatial variability. Without such sensitivity testing, it is unclear whether the reported hemispheric coordination of GPP patterns reflects a dynamically robust signal or is partially dependent on the characteristics of the NIRv product.

Response: We thank the reviewer for this valuable comment, which helps improve the robustness of our results. In response, we have incorporated 3 additional GPP datasets for supplementary validation:

i). The GLASS project provides GPP (GLASS GPP) products with a long period of coverage (1982–2018) (Liang et al., 2021). The algorithm of this GPP product was proposed by Yuan et al., using the eddy covariance-light use efficiency (EC-LUE) model, which integrates 8 LUE models used extensively worldwide (Yuan et al., 2007, 2010).

ii). Global monthly GPP from an improved LUE Model during 1982-2016 (Madani et al., 2017). This dataset (LUE GPP) was improved with optimized spatially and temporally explicit LUE values derived from selected FLUXNET tower site data. Global gridded long-term daily GPP was derived using the optimized LUE, Global Inventory Modeling and Mapping Studies (GIMMS3g) canopy fraction of photosynthetically active radiation (FPAR), and Modern-Era Retrospective analysis for Research and

Response Letter

Applications, Version 2, (MERRA-2) meteorological information. These data will improve satellite-based estimation and understanding of GPP using a refined LUE model framework.

iii). LRF GPP, a GPP product proposed by Tagesson et al. (2021). It is a new method to estimate the ecosystem-level physiological approach of GPP using the asymptotic LRF between GPP and incoming photosynthetically active radiation. This GPP dataset ranging from 1982 to 2016.

We specifically selected these datasets because they provide longer temporal coverage and spatially gridded information. Most other GPP products only cover a relatively short period after 2000, whereas the curvature of the Northern Hemisphere westerly jet, which is central to our study, changed around 2000. Using short-term post-2000 data alone would not fully capture the relationship between GPP and jet curvature. Additionally, although FLUXNET observations are invaluable, their spatial coverage is sparse in many of our hotspot regions, and the record lengths are limited. The three datasets we selected include LUE GPP, which extensively utilized FLUXNET data for gap-filling and optimization, while GLASS GPP and LRF GPP have also been thoroughly validated in previous studies, demonstrating good agreement with observational data and strong applicability.

Our analysis shows that while absolute values and trends differ across GPP products due to variations in retrieval and calculation methods, the overall spatial pattern of GPP associated with westerly jet curvature in our hotspot regions (R1–R5) remains unchanged. This indicates that the influence of westerly jet curvature on the large-scale spatial GPP patterns in mid- to high-latitude Northern Hemisphere hotspots is a robust mechanism, independent of the specific GPP dataset used. In regions outside the hotspots considered in this study, the responses of different GPP products to westerly jet curvature exhibit slight spatial differences (e.g., in eastern China and tropical Africa), but the overall pattern remains highly consistent, with spatial correlation coefficients between products that are significant at the 99% confidence level.

We have added the following description in the "Reanalysis and Observation Datasets" section (Lines 357 to 367): **To evaluate whether the robustness of GPP responses to westerly jet curvature is affected by dataset differences, we additionally used three GPP products for validation (the additional GPP data are used Figure S4): (1) GLASS GPP, covering 1982–2018 at 0.05° resolution, derived using an eddy covariance–light use efficiency (EC-LUE) model that integrates multiple globally used LUE models (Liang et al., 2021; Yuan et al., 2007, 2010). (2) LUE GPP, a global monthly GPP product covering 1982–2016 at 8 km resolution, estimated with an optimized LUE model based on FLUXNET data, combined with GIMMS3g FPAR and MERRA-2 meteorology data (Madani et al., 2017). (3) LRF GPP, spanning 1982–2016 at 0.05° resolution, estimated using an asymptotic light response function between GPP and incoming photosynthetically active radiation data (Tagesson et al., 2021). The results show that while minor spatial differences exist among datasets (e.g., in tropical Africa and eastern China),**

Response Letter

the overall spatial pattern in mid- to high-latitude regions is highly consistent, particularly within the hotspots considered in this study.

Figure S4. Relationships between summer westerly patterns in the Northern Hemisphere and gross primary productivity (GPP) based on different products. Spatial distributions of correlation coefficients between local summer GPP and Northern Hemisphere overall curvature (1982–2018) using (a) GLASS GPP, (b) LUE GPP, and (c) LRF GPP products. Only correlations significant at the 95% confidence level are shown.

Reference

Liang, S. et al. The Global Land Surface Satellite (GLASS) Product Suite. *Bulletin of the American Meteorological Society* 102, E323–E337 (2021).

Response Letter

Madani, N., Kimball, J. S. & Running, S. W. Improving Global Gross Primary Productivity Estimates by Computing Optimum Light Use Efficiencies Using Flux Tower Data. *JGR Biogeosciences* 122, 2939–2951 (2017).

Tagesson, T. et al. A physiology-based Earth observation model indicates stagnation in the global gross primary production during recent decades. *Global Change Biology* 27, 836–854 (2021).

Yuan, W. et al. Global estimates of evapotranspiration and gross primary production based on MODIS and global meteorology data. *Remote Sensing of Environment* 114, 1416–1431 (2010).

Yuan, W. et al. Deriving a light use efficiency model from eddy covariance flux data for predicting daily gross primary production across biomes. *Agricultural and Forest Meteorology* 143, 189–207 (2007).

4. Figure 2 highlights regions (specifically R2, R5, and also Southeast Asia) that overlap substantially with cropland-dominated areas. These regions are agricultural managed systems, where human influence, such as irrigation, fertilisation, crop rotation, and land-use practices exert strong control over GPP variability. In such landscapes, anthropogenic management can significantly modulate or even override climate-driven constraints on productivity. While large-scale westerly jet curvature shifts may influence regional hydroclimate (e.g., precipitation patterns, temperature extremes, or radiation anomalies), their direct control over GPP in managed croplands may be weaker or strongly mediated by human interventions. For example, irrigation can buffer precipitation deficits, and fertilizer inputs can enhance photosynthetic capacity independent of circulation-induced variability. Given that the manuscript emphasizes hemispheric-scale coordination of GPP patterns driven by jet curvature changes, it would be important to clarify whether the reported relationships hold consistently across different land-cover types, particularly between natural ecosystems and intensively managed croplands.

Response: We thank the reviewer for this insightful comment. To address it, we used the historical LUH2 dataset (<https://luh.umd.edu/index.shtml>) and applied bilinear interpolation to obtain land-use fractions at the same spatial resolution as the GPP data. Each grid cell was then categorized by land-cover type, distinguishing managed land (croplands and urban areas) from natural land (primary and secondary forests), and mean land-cover fractions were calculated over the period 1982–2015 (as limited by the historical LUH2 dataset). Using this classification, we refitted the correlations in the five hotspot regions highlighted in Figure 2c, separating managed and natural lands. Consistent with expectations, the relationship between GPP and westerly jet curvature is modulated by land-use type, with stronger correlations observed in grid cells containing a higher fraction of natural land. These findings indicate that human activities can attenuate the climate-driven influence of westerly jet curvature, so that its impact on GPP is most evident in largely natural landscapes. In addition, we focus on extratropical regions because the westerly jet exerts its direct influence primarily outside the tropics.

Response Letter

In contrast, the effects of the jet on tropical areas, such as Southeast Asia, generally require more complex teleconnection mechanisms to be transmitted. We have added the following sentences to encompass these results (Lines 164 to 172): Given that anthropogenic land management can modulate or alleviate climate-driven constraints on productivity, we evaluated how land-use type affects the strength of the correlation between westerly jet curvature and GPP in the five hotspot regions shown in Figure 2c. Each grid cell was classified as either managed land (croplands and urban areas) or natural land (primary and secondary forests) using LUH2 data, and the absolute correlation ($|r|$) between westerly jet curvature and GPP was calculated for each cell. The results show that the influence of large-scale westerly jet curvature on GPP is weaker over managed lands (Figure S3a), likely because human interventions such as irrigation, fertilization, and crop rotation buffer or override climate-driven variability^{54,55}. In contrast, correlations are stronger in natural ecosystems (Figure S3b), where productivity is more directly constrained by climate.

We have also added a description of the LUH2 dataset in the method section (Lines 368 to 372): LUH2 provides annually resolved, spatially explicit information on global land-use states and transitions at a horizontal resolution of 0.25°, including cropland, urban, forest, and other land-cover types (Hurtt et al., 2011; 2020). This dataset enables us to distinguish intensively managed croplands from natural ecosystems in our analyses and to better account for potential human influences on GPP variability. We used LUH2 to calculate mean land-use fractions over the historical period 1982–2015.

References

- Hurtt, G. C. et al. Harmonization of land-use scenarios for the period 1500–2100: 600 years of global gridded annual land-use transitions, wood harvest, and resulting secondary lands. *Climatic Change* 109, 117–161 (2011).
- Hurtt, G. C. et al. Harmonization of global land use change and management for the period 850–2100 (LUH2) for CMIP6. *Geosci. Model Dev.* 13, 5425–5464 (2020).
- Yang, X. et al. Diversifying crop rotation increases food production, reduces net greenhouse gas emissions and improves soil health. *Nat Commun* 15, 198 (2024).
- Yang, Y. et al. Sustainable irrigation and climate feedbacks. *Nat Food* 4, 654–663 (2023).

Response Letter

Figure S3. Influence of land-use type on the relationship between westerly jet curvature and GPP. Joint distributions of (a) managed land fraction (%) and (b) natural land fraction (%) with the absolute correlation between westerly jet curvature and GPP ($|r|$). Overlaid histograms show the number of grid cells in each $|r|$ bin, and black dashed lines indicate linear regression fits.

5. Line 189-225: While the CESM experiments suggest GHG dominance over aerosols and biomass burning in the modelled response of Northern Hemisphere westerly curvature changes, the physical pathway linking GHG forcing to these curvature changes requires clearer articulation. Explicitly connecting the diagnosed curvature changes to established dynamical mechanisms would strengthen confidence in the attribution and clarify how this relates to existing westerly jet-shift drives literature, e.g. Arctic warming (Cheng et al 2025), tropical warming (Woollings et al 2023).

Response: We thank the reviewer for this insightful comment. To clarify the physical pathways linking GHG forcing to Northern Hemisphere westerly curvature changes, we have added the following explanation in the revised manuscript (Lines 249 to 256): The observed patterns can be understood through established mechanisms, whereby GHG forcing modifies the meridional temperature gradient between the Arctic and mid-latitudes, thereby modulating the baroclinicity and wave propagation of the jet stream⁶²⁻⁶⁴. These changes in thermal-structure enhance jet meandering and amplify curvature anomalies, reflecting the link between Arctic amplification and shifts in mid-latitude circulation^{40,65}. In addition, tropical warming provides an indirect pathway for influencing mid-latitude westerlies by reshaping upper-tropospheric waveguides and Rossby wave propagation, thereby complementing the effects of high-latitude warming on jet curvature^{42,66}.

References

Cheng, L. et al. Westerly jet waviness modulates mid-latitude hydroclimate variability. *Nat Commun* 16, 10928 (2025).

Response Letter

Woollings, T., Drouard, M., O'Reilly, C. H., Sexton, D. M. H. & McSweeney, C. Trends in the atmospheric jet streams are emerging in observations and could be linked to tropical warming. *Commun Earth Environ* 4, 125 (2023).

Coumou, D., Lehmann, J. & Beckmann, J. The weakening summer circulation in the Northern Hemisphere mid-latitudes. *Science* 348, 324–327 (2015).

Breul, P., Ceppi, P., Simpson, I. R. & Woollings, T. Seasonal and regional jet stream changes and drivers. *Nat Rev Earth Environ* 6, 824–842 (2025).

Chen, G., Zhang, P. & Lu, J. Sensitivity of the Latitude of the Westerly Jet Stream to Climate Forcing. *Geophysical Research Letters* 47, e2019GL086563 (2020).

Iles, C. E., Samset, B. H. & Lund, M. T. How polar-midlatitude atmospheric teleconnections depend on regional sea ice fraction and global warming level. *Earth Syst. Dynam.* 16, 2253–2272 (2025).

Response Letter

General Response to the Reviewers' Comments

Dear Editor and Referees,

We would like to express our deepest gratitude to the reviewer for the exceptionally rigorous and constructive feedback. Your professional insights and the significant time you invested in reviewing our manuscript have been instrumental in refining our work. We have also tracked and highlighted the major revisions made in response to each comment in the revised manuscript. For further details, please refer to our point-by-point responses. The reviewers' comments are shown in **black**, our responses are provided in **blue**, and the corresponding changes in the manuscript are highlighted in **red**.

Response Letter

REVIEWERS' COMMENTS:

Reviewer #2

The authors have done a significant amount of work to improve the manuscript, and it now looks much better. I have a few minor comments:

1. While the authors attempt to decompose the influence of westerly curvature into “direct” and “indirect” components, the physical interpretation of these pathways remains somewhat unclear. In Text S1, “The information flow from westerly curvature to GPP is 0.26, indicating that westerly curvature is an important driver and direct influencing factor of GPP variability.” The direct pathway (westerly curvature \rightarrow GPP) assumes the existence of a mechanism independent of hydroclimate variables, which is not fully explained. It may be helpful to briefly discuss the physical plausibility of such a direct pathway independent of precipitation and temperature, or to adopt more cautious wording, rather than implying a strictly independent causal mechanism, especially when the manuscript acknowledges the difficulty of isolating the influence of individual variables in a coupled earth system.

Response: Thank you for this insightful comment. We agree with your concern and have revised the relevant text to improve the clarity and interpretation. As you rightly pointed out, a strict separation between “direct” and “indirect” pathways may lead to ambiguity, particularly when attempting to attribute an independent physical mechanism for the influence of westerly jet curvature on GPP. In a coupled Earth system, it is indeed difficult to fully disentangle such pathways, and any apparent “direct” effect may still implicitly reflect the combined influence of multiple interacting processes. Based on its definition, the information-flow-based causal metrics quantify directed dependencies embedded in the system dynamics. They should be interpreted as representing net effects within a multivariate system, potentially integrating both accounted and unaccounted pathways. In the revised manuscript, we have therefore adopted more cautious wording and reinterpreted the results (Lins 63-92 in Supplementary Information): In this causal network (Extended Figure 7), the role of westerly curvature can be interpreted as an upstream dynamical regulator within a coupled “circulation dynamics–surface (hydro-) climate factors–GPP response” cascade. Its influence on GPP is not realized through a single pathway, but instead emerges from multiple interacting and nonlinearly coupled transmission routes. Structurally, westerly curvature acts as a diagnostic metric of Rossby wave morphology. By modulating jet stream meandering and ridge–trough configurations, it is significantly associated with the 200-hPa geopotential height ($T_{\text{westerly curvature} \rightarrow z_{200}}=0.24$) and meridional wind fields ($T_{\text{westerly curvature} \rightarrow v_{200}}=0.19$). This indicates that geometric changes in the mid- to high-latitude circulation induce a systematic dynamical reorganization, leading to enhanced waviness of the large-scale flow and strengthened meridional transport. Building on this dynamical adjustment, anomalies in geopotential height and meridional wind jointly regulate surface hydroclimatic conditions, thereby reshaping regional energy balance and

Response Letter

moisture transport pathways. At the climate response level, circulation anomalies exert a pronounced influence on two key ecological limiting factors: maximum surface air temperature ($T_{z_{200} \rightarrow T_{max}}=0.20$) and precipitation ($T_{z_{200} \rightarrow \text{Precipitation}}=0.14$; $T_{V_{200} \rightarrow \text{Precipitation}}=0.17$). Importantly, temperature and precipitation do not respond independently; instead, they exhibit coupled responses under the same large-scale circulation background ($T_{T_{max} \rightarrow \text{Precipitation}}=0.36$). At the GPP response level, GPP is jointly constrained by both precipitation and temperature, although the precipitation limitation is more dominant ($T_{\text{Precipitation} \rightarrow \text{GPP}}=0.33$). Meanwhile, temperature plays a secondary regulatory role by influencing evapotranspiration demand and other physiological requirements for vegetation growth ($T_{T_{max} \rightarrow \text{GPP}}=0.23$).

Therefore, the whole structure of the causal network reveals a clear yet highly coupled propagation chain: west wind curvature, as an upstream dynamical modulator, reshapes the upper-level circulation, subsequently affects key (hydro-) climatic variables such as temperature and precipitation, and then regulates variations in GPP ($T_{\text{westerly curvature} \rightarrow \text{GPP}}=0.26$). More importantly, the inferred network reflects not merely a linear cascade, but a multilevel system characterized by feedback coupling. West wind curvature not only influences downstream variables through geopotential height and meridional wind, but these climatic variables also interact with each other. As a result, changes in thermal and hydrological conditions are not simply additive; rather, they are systematically amplified or suppressed through dynamic–thermodynamic coupling, highlighting the presence of complex nonlinear interactions within the climate system.

We also revise the related description in the main text (Lines 226-232):

Causal analysis further reveals that variations in westerly curvature may exert broad-scale influences on GPP through circulation adjustments; however, these effects are also mediated and amplified by local temperature and precipitation, which exert more immediate control over GPP. A detailed description of the causal analysis framework used to identify and quantify the direct and indirect influences of westerly curvature, circulation factors, and surface climate variables on GPP is provided in the Supporting Information (Text S1 and Extended Figure 7).

2. Does the Liang–Kleeman information flow approach applied here require the underlying time series to be stationary, particularly given the apparent reversal in westerly jet curvature around the year 2000?

Response: The Liang–Kleeman information flow framework does not require strict stationarity of the underlying time series, as it is derived from a general stochastic dynamical system rather than relying on traditional time series assumptions such as those in Granger causality. Because the method does not impose a specific model structure (Liang, 2014), it avoids the stationarity constraints inherent in autoregressive approaches. It has also been successfully applied to climate systems characterized by

Response Letter

strong nonstationarity and regime shifts, including ENSO variability (Liang, 2014), and has been widely used in subsequent studies of nonstationary climate dynamics (e.g., Stips et al., 2016; Sun et al., 2025). Therefore, the reversal observed around 2000 does not invalidate the use of Liang-Kleeman framework in principle. We have revised the Supplementary Information to further emphasize the broad applicability of the information flow approach (Lines 39-42 in Supplementary Information):

Unlike Granger causality, which is primarily based on predictive capability within linear statistical models, the information flow approach is better suited for characterizing complex nonlinear interactions and stochastic dynamics in climate systems (Stips et al., 2016; Sun et al., 2025).

In this study, although a reversal in the westerly jet curvature appears around 2000, it does not fundamentally affect the estimation of information flow. To further ensure robustness, we tested the sensitivity of the results using detrended data (see figure below). For the identified key network nodes, although the information flow exhibits numerical fluctuations, it remains significantly distinct from that of non-key nodes. These results indicate that the inferred causal network is largely insensitive to the presence or absence of long-term trends.

Figure for Comment 2. Standardized information flow matrix among westerly curvature, circulation factors including 200 hPa geopotential height, zonal (U200) and meridional (V200) winds, local climate variables including surface maximum temperature (Tmax) and precipitation, and gross primary production (GPP) from 1982 to 2018 (n=37). Panels (a) and (b) show results based on detrended and original data, respectively. The color of each circle reflects the mean value across the five sectors

Response Letter

(Eastern Europe, Central Asia, East Asia, Western North America, and Central North America), and the size corresponds to the standard deviation among the sectors.

3. Like other causal discovery approaches, the Liang–Kleeman Information Flow also depends on sample size. It would be helpful to report the sample size, as in data-driven approaches with small datasets, we cannot confidently conclude that the inferred causal network represents true causal relationships.

Response: Thank you for your helpful suggestion. We have now added the sample size for each time series in the revised manuscript to clarify the data basis of the analysis. In the captions of Extended Figure 7, we clarified that the time series spans 1982 to 2018 ($n = 37$). The uncertainty is calculated as the standard deviation across five sectors (Eastern Europe, Central Asia, East Asia, Western North America, and Central North America).

4. The sign of the relationship is consistent across different GPP datasets; however, the strength and the number of statistically significant pixels vary, despite all datasets covering the same period. This is particularly evident for the LUE-based GPP product, which shows relatively few statistically significant pixels. This difference could be more clearly illustrated by identifying or quantifying consistent and inconsistent pixels across datasets. A brief comment (1–2 lines) on uncertainties associated with different GPP products would also be helpful.

Response: We agree with your comment. The different GPP products show consistent large-scale spatial patterns in the relationship; however, the number of pixels passing the significance test varies among datasets. For the hotspot regions of interest in this study, this discrepancy is relatively small. Using the NIRv GPP dataset as the reference, we calculated the percentage of consistently significant pixels within the five hotspot regions (R1–R5). In addition, we added a short discussion on the associated uncertainties in the revised manuscript (Lines 376–385): **The results show that while minor spatial differences exist among datasets (e.g., in tropical Africa and eastern China), the overall spatial pattern in mid- to high-latitude regions is highly consistent, particularly within the hotspots considered in this study (Figure S4). In addition, except for the LUE GPP, the other GPP products show a high degree of consistency in the spatial distribution of statistically significant responses to westerly curvature, reaching an average agreement of 78.3% within the hotspot regions (using the NIRv GPP dataset as a reference). Although the LUE GPP exhibits a similar spatial pattern, it yields fewer statistically significant pixels (Figure S4b). These differences likely reflect structural uncertainties among GPP products arising from different model assumptions, input datasets, and parameterizations,**

Response Letter

particularly in light-use-efficiency-based estimates. Overall, the main conclusions of this study are robust and not sensitive to the selection of GPP dataset.

5. Lines 68–69: The cited reference uses NDVI, and NDVI and GPP may diverge. It would be useful to explicitly mention the variable used in the citation to avoid confusion. For example: “Recent land vegetation growth, observed through satellite-derived Normalised Difference Vegetation Index (NDVI), is regulated by natural climate variability such as the El Niño–Southern Oscillation (ENSO) and anthropogenic climate change.”

Response: Thank you for pointing out this issue. We have corrected it accordingly (Lines 68-71): Using satellite-derived Normalized Difference Vegetation Index, a recent study has shown that terrestrial vegetation greening is regulated by both natural climate variability, such as the El Niño–Southern Oscillation (ENSO), and anthropogenic climate change.

6. Figure 1 caption currently explains only panels (a) and (b). It would be helpful to include brief descriptions of panels (c–k) to improve clarity for the reader.

Response: Thanks for your comment. We have revised the caption of Fig. 1, with additional clarification of c–k (the relevant parts have been highlighted in bold): **Figure 1. Summer westerly curvature in the Northern Hemisphere and its changes during 1979–2023. (a) Mean summer (June–July–August) curvature (black line, 10^{-5} degree $^{-1}$) of the westerly jet axis at 200hPa and its trend (shading, 10^{-5} degree $^{-1}$ year $^{-1}$) during 1979–2023 based on ERA5 daily data. Hatched areas indicate significant trends at the 95% confidence level. (b) Summer westerly curvature anomalies (relative to the 1979–2023 mean) during 1979–1999, (blue line) and 2000–2023 (red line). The Northern Hemisphere is divided into eight longitudinal regions: Western Europe (WE), Eastern Europe (EE), Central Asia (CA), Eastern Asia (EA), Pacific Ocean (PA), Western North America (WN), Central North America (CN), and Eastern North America (EN). The basemaps in (a) and (b) are shown only for reference of the longitude range on the x-axis. The y-axis represents curvature and is unrelated to latitude. The westerly jet axis detection is performed within the Northern Hemisphere (0–90° N). A positive curvature indicates a northern ridge, while a negative curvature is for a southward trough. (c–j) Time series of summer mean westerly curvature κ (solid lines) and summer mean absolute westerly curvature $|\kappa|$ (dashed lines) for (c) WE, (d) EE, (e) CA, (f) EA, (g) PA, (h) WN, (i) CN, and (j) EN. (k) Trends of summer mean westerly curvature (red bars) and summer mean absolute westerly curvature (blue bars) across the eight sub-regions. Asterisks indicate statistical significance at the 95% confidence level.**

Response Letter

7. Line 225: Consider replacing “ecosystem productivity” with “vegetation productivity” or simply “GPP” for clarity.

Response: We replaced it with “GPP”.

8. Lines 284–286: The statement that “the difference between GPP trends under greenhouse gas and non-greenhouse gas forcing closely matches the observed spatial pattern” may need reconsideration. In Western North America, observations indicate a declining GPP trend, whereas the GHG-minus-non-GHG difference suggests an increase, indicating a mismatch in this region.

Response: Thank you for your careful observation. We have revised this part to provide a clearer and more complete explanation. Overall, although westerly circulation can systematically shape the large-scale spatial patterns of GPP across the Northern Hemisphere, its impacts must be interpreted in the context of regional hydroclimatic differences.

In our observation-based analysis, westerly curvature in Western North America (WN) shows a weak increasing trend ($2.06 \text{ degree}^{-1} \text{ yr}^{-1}$), but it is not statistically significant (Fig. 1k). In this region, GPP exhibits a negative relationship with the overall curvature (Fig. 2c). As shown in Extended Figs. 6g and 6h, higher curvature is associated with a slight shift of T_{max} toward higher values, whereas precipitation shows a more visible shift toward lower values. This reduction in precipitation leads to a negative GPP response (Extended Fig. 5h). In contrast, under future greenhouse gas (GHG) forcing, curvature increases significantly in this region (Fig. 4d and g), accompanied by a marked rise in T_{max} (Extended Fig. 9d). Importantly, T_{max} remains below the inflection point of the GPP response during our study period (Extended Fig. 5g), thereby favoring increased GPP. Meanwhile, curvature-induced anticyclonic anomalies tend to suppress precipitation, whereas warming enhances atmospheric moisture content and partially offsets this effect. As a result, future changes in precipitation remain relatively small (Extended Fig. 10d). Taken together, these results suggest that during the observational period, the negative GPP response to curvature in WN is primarily driven by precipitation deficits, whereas in the future, the stronger increase in T_{max} becomes the key factor controlling GPP. This leads to a mismatch between observed trends and the GHG-minus-non-GHG signal in this region. We have revised the manuscript to clarify that the influence of hemispheric-scale westerly curvature on GPP is strongly modulated by regional hydroclimatic conditions, and therefore may exhibit regionally contrasting responses (Lines 286-300): Overall, the spatial pattern of the GPP trend difference between greenhouse gas and non-greenhouse gas forcing broadly resembles the observed pattern across most regions, although discrepancies remain in western North America. In this region, the relationship between westerly curvature and GPP exhibits a distinct regional behavior. During the observational period, the negative GPP response to curvature is primarily associated with reduced precipitation (Extended Figure 6h), which plays a dominant limiting role. In contrast, under greenhouse gas forcing, the projected increase

Response Letter

in curvature is accompanied by a stronger rise in Tmax (Extended Figure 9d), which remains below the inflection point of the GPP response and thus favors vegetation productivity (Extended Figure 5g). Meanwhile, changes in precipitation are comparatively weak due to the competing effects of circulation-induced subsidence and thermodynamically enhanced moisture availability. As a result, future changes in GPP are more strongly associated with temperature increases than with precipitation changes, leading to a divergence between observed trends and the GHG-minus-non-GHG signal in this region. This indicates that, although westerly circulation can systematically shape the large-scale spatial patterns of GPP across the Northern Hemisphere, its impacts must be interpreted in the context of regional hydroclimatic differences.